# Role of oxidation of excitation-contraction coupling machinery in age-dependent loss of muscle function in *Caenorhabditis elegans*

Haikel Dridi[1], Frances Forrester[1], Alisa Umanskaya[1], Wenjun Xie[1], Steven Reiken[1], Alain Lacampagne[2,3], Andrew Marks[1]*

[1]Department of Physiology and Cellular Biophysics, The Clyde and Helen Wu Center for Molecular Cardiology, New York, United States; [2]PhyMedExp, Montpellier University, INSERM, CNRS, CHRU Montpellier, Montpellier, France; [3]Medical Intensive Care Unit, Montpellier University and Montpellier University Health Care Center, Montpellier, France

**Abstract** Age-dependent loss of body wall muscle function and impaired locomotion occur within 2 weeks in *Caenorhabditis elegans (C. elegans)*; however, the underlying mechanism has not been fully elucidated. In humans, age-dependent loss of muscle function occurs at about 80 years of age and has been linked to dysfunction of ryanodine receptor (RyR)/intracellular calcium ($Ca^{2+}$) release channels on the sarcoplasmic reticulum (SR). Mammalian skeletal muscle RyR1 channels undergo age-related remodeling due to oxidative overload, leading to loss of the stabilizing subunit calstabin1 (FKBP12) from the channel macromolecular complex. This destabilizes the closed state of the channel resulting in intracellular $Ca^{2+}$ leak, reduced muscle function, and impaired exercise capacity. We now show that the *C. elegans* RyR homolog, *UNC-68*, exhibits a remarkable degree of evolutionary conservation with mammalian RyR channels and similar age-dependent dysfunction. Like RyR1 in mammals, *UNC-68* encodes a protein that comprises a macromolecular complex which includes the calstabin1 homolog FKB-2 and is immunoreactive with antibodies raised against the RyR1 complex. Furthermore, as in aged mammals, *UNC-68* is oxidized and depleted of FKB-2 in an age-dependent manner, resulting in 'leaky' channels, depleted SR $Ca^{2+}$ stores, reduced body wall muscle $Ca^{2+}$ transients, and age-dependent muscle weakness. FKB-2 (*ok3007*)-deficient worms exhibit reduced exercise capacity. Pharmacologically induced oxidization of *UNC-68* and depletion of FKB-2 from the channel independently caused reduced body wall muscle $Ca^{2+}$ transients. Preventing FKB-2 depletion from the *UNC-68* macromolecular complex using the Rycal drug S107 improved muscle $Ca^{2+}$ transients and function. Taken together, these data suggest that *UNC-68* oxidation plays a role in age-dependent loss of muscle function. Remarkably, this age-dependent loss of muscle function induced by oxidative overload, which takes ~2 years in mice and ~80 years in humans, occurs in less than 2–3 weeks in *C. elegans*, suggesting that reduced antioxidant capacity may contribute to the differences in lifespan among species.

*For correspondence:
arm42@cumc.columbia.edu

## Editor's evaluation

This manuscript will appeal to all with an interest in comparative physiology and the molecular biology of age-associated changes in muscle function. The authors draw parallels between aging skeletal muscle in humans and *C. elegans*, with evidence in support of age-dependent oxidation of the *C. elegans* ryanodine receptor ortholog, UNC-68, causing loss of the calstabin ortholog, FKB-2. This in turn results in calcium leak, reduced body wall calcium transients and muscle weakness,

changes that are similar to those that occur in aging human skeletal muscle despite the dramatic differences in the lifespan of the two organisms.

## Introduction

Approximately 50% of humans over the age of 80 develop muscle weakness, which contributes to falls and hip fractures, a leading cause of mortality in the elderly (*Marzetti and Leeuwenburgh, 2006*; *Ganz et al., 2007*; *Santulli et al., 2013*). Strikingly, despite an approximately 2000-fold difference in the lifespans of humans and *Caenorhabditis elegans* (*Herndon et al., 2002*; *Ljubuncic and Reznick, 2009*), both exhibit oxidative overload induced age-dependent reductions in muscle function and motor activity that ultimately contribute to senescence and death. Due to its short lifespan and well-characterized genome, *C. elegans* has been used as a model to study the genetics of aging and lifespan determination (*Guarente and Kenyon, 2000*; *Kenyon, 2010*), including the age-dependent decline in locomotion (*Herndon et al., 2002*; *Hsu et al., 2009*). Age-dependent reduction in locomotion in *C. elegans* has been attributed to degeneration of the nervous system (*Liu et al., 2013*) and the body wall musculature (*Kirkwood, 2013*). Here, we investigated the role of the ryanodine receptor (RyR)/intracellular calcium (Ca$^{2+}$) release channel homolog, *UNC-68,* in age-dependent loss of muscle function in *C. elegans*.

Mammalian RyR1 is the major intracellular Ca$^{2+}$ release channel in skeletal muscle required for excitation-contraction (E-C) coupling (*Zalk et al., 2015*). In mammals, peak intracellular Ca$^{2+}$ transients evoked by sarcolemmal depolarization decrease with age (*Gonzalez et al., 2003*), and this decrease is associated with a reduced SR Ca$^{2+}$ release (*Jiménez-Moreno et al., 2008*) that directly determines the force production of skeletal muscle. Our group has shown that a mechanism underlying age-dependent loss of muscle function is RyR1 channel oxidation which depletes the channel complex of the stabilizing subunit calstabin1 (calcium channel stabilizing binding protein type 1, or FKBP12), resulting in intracellular Ca$^{2+}$ leak and muscle weakness (*Andersson et al., 2011*; *Umanskaya et al., 2014*). RyR1 is a macromolecular complex comprised of homotetramers of four ~565 kDa RyR monomers (; *Zalk et al., 2007*). Cyclic AMP (cAMP)-dependent protein kinase A (PKA) (*Marx et al., 2000*), protein phosphatase 1 (*Kass et al., 2003*), phosphodiesterase PDE4D3 (*Lehnart et al., 2005*), Ca$^{2+}$-dependent calmodulin kinase II (CaMKII) (*Currie et al., 2004*; *Kushnir et al., 2010*), and calstabin1 (*Bellinger et al., 2008*) are components of the RyR1 macromolecular complex (*Santulli and Marks, 2015*). Calstabin1 is part of the RyR1 complex in skeletal muscle, and calstabin2 (FKBP12.6) is part of the RyR2 complex in cardiac muscle (*Santulli et al., 2017*). Calstabins are immunophilins (*Marks, 1996*) with peptidyl-prolyl isomerase; however, this enzymatic activity does not play a role in regulating RyR channels and rather they stabilize the closed state of RyRs and prevent a Ca$^{2+}$ leak via the channel (*Marx et al., 2000*; *Brillantes et al., 1994*).

RyR belongs to a small family of large intracellular Ca$^{2+}$ release channels, the only other member being the inositol 1,4,5-triphosphate receptor (IP$_3$R) (*Harnick et al., 1995*; *Jayaraman et al., 1995*; *Jayaraman and Marks, 2000*). RyR may have evolved from IP$_3$R-B, which encoded an IP$_3$R-like channel that could not bind IP$_3$ and was replaced by RyR at the Holozoa clade (*Alzayady et al., 2015*). Invertebrates have one gene for each of RyR and IP$_3$R, while vertebrates have three (RyR1-3 and IP$_3$R1-3). RyRs and IP3Rs are intracellular Ca$^{2+}$ release channels on the SR/ER and are tetramers that along with associated proteins comprise the largest known ion channel macromolecular complexes (*Marx et al., 2000*; *DeSouza et al., 2002*). Defects in Ca$^{2+}$ signaling linked to stress-induced remodeling that results in leaky RyR channels have been implicated in heart failure (*Dridi et al., 2020c*; *Marks, 2003*), cardiac arrhythmias (*Dridi et al., 2020c*; *Lehnart et al., 2006*; *Lehnart et al., 2004*; *Vest et al., 2005*; *Wehrens et al., 2003*), diabetes (*Santulli et al., 2015*), muscle weakness (*Kushnir et al., 2020*; *Dridi et al., 2020b*; *Matecki et al., 2016*; *Dridi et al., 2020d*), and neurodegenerative disorders (*Dridi et al., 2020b*; *Lacampagne et al., 2017*; *Liu et al., 2012*).

RyR has evolved unique SPRY domains (*des Georges et al., 2016*) that are absent in IP$_3$R, one of which (SPRY2) allows RyR1 to directly interact with the L-type calcium channel (Cav1.1) in mammalian skeletal muscle (*Cui et al., 2009*). This interaction couples excitation of the sarcolemma to muscle contraction to overcome the dependence on extracellular Ca$^{2+}$. RyR1 is remarkably well conserved, suggesting that independence from extracellular Ca$^{2+}$ evolved to support locomotion in higher organisms.

*UNC-68* is the RyR gene homolog in the *C. elegans* genome (*Maryon et al., 1996*). Worms lacking both exon 1.1 and promoter1 (*Marques et al., 2020*), and *UNC-68 (e540)* null mutants exhibit severely defective swimming behavior and locomotive characterized by the 'unc', or 'unco-ordinated' phenotype (*Brenner, 1974*). Treatment with ryanodine, a chemical ligand of RyR, induces contractile paralysis in wild-type (WT) *C. elegans*, whereas *UNC-68 (e540)* null mutants are unaffected by ryanodine (*Maryon et al., 1996*; *Brenner, 1974*; *Sakube et al., 1997*). $Ca^{2+}$ transients triggered by action potentials in *C. elegans* body wall muscles require *UNC-68*.

We previously reported that in aged mice (2 years old equivalent to ~80-year old humans) RyR1 oxidation depletes calstabin1 from the channels and renders them leaky to $Ca^{2+}$, which contributes to the loss of muscle function and impaired muscle strength (*Umanskaya et al., 2014*). In the present study, we show that *UNC-68* is comprised of a macromolecular complex that is remarkably conserved compared to RyR1 and includes the channel-stabilizing subunit, *FKB-2*. Like calstabin, *FKB-2* regulates *UNC-68* by directly associating with the channel. Similar to what we previously observed in mice (*Andersson et al., 2011*), we found age-dependent oxidation of *UNC-68* which causes depletion of *FKB-2* from the *UNC-68* channel complex, and reduces $Ca^{2+}$ transients in aged nematodes. This aging phenotype was accelerated in *FKB-2 (ok3007)* worms, an *FKB-2* deletion mutant that results in leaky *UNC-68*. Competing *FKB-2* from *UNC-68* with rapamycin or FK506 (*Timerman et al., 1993*) resulted in reduced body wall muscle $Ca^{2+}$ transients and defective locomotion. Conversely, pharmacological and genetic oxidation of *UNC-68* with the reactive oxygen species (ROS)-generating drug paraquat (*Lee et al., 2003*) caused *FKB-2* dissociation from the channel and reduced contraction-associated $Ca^{2+}$ transients. Reassociating *FKB-2* with *UNC-68* using the RyR-stabilizing drug S107 improved $Ca^{2+}$ transients and locomotion in aged nematodes. We have recently reported the binding site for S107 and its second generation Rycal, ARM210, using cryogenic electron microscopy (*Melville et al., 2022*). The compound binds in a cleft in the cytosolic shell and prevents a remodeled RyR channel from sitting in a 'primed state' sensitive to activation (*Melville et al., 2022*; Miotto et al., in-revision Science Advances 2022). A clinical trial using ARM210 to fix the leak in RyR1 channels is currently underway at the NIH (NCT04141670).

Our study provides an underlying mechanism for age-dependent loss of muscle function in *C. elegans* including progressive oxidation of *UNC-68*, which depletes the stabilizing binding protein, FKB-2 and, renders the channel leaky within 2 weeks compared to 2 years in mice and 80 years in humans and a potential therapy.

## Results

### Conserved evolution and architecture of UNC-68

Phylogenic analysis of RyR and FKBP among species reveals remarkable evolutionary conservation (*Figure 1A–B*). *UNC-68*, the *C. elegans* intracellular calcium release channel, shares ~40% homology with the human RyR1 (*Figure 1C*). *C. elegans FKB-2* has ~60% sequence identity with the skeletal muscle isoform calstabin1 (FKBP12) (*Figure 1D*). Based on these observations, we hypothesized that in *C. elegans, UNC-68* comprises a macromolecular complex, similar to that of mammalian RyRs. To test this hypothesis, lysates were prepared from populations of freeze-cracked WT *C. elegans,* and *UNC-68* was immunoprecipitated using mammalian anti-RyR antibody (5029) as previously described (*Kushnir et al., 2018*). The immunoprecipitates were immunoblotted to detect *UNC-68,* as well as other components of the RyR macromolecular complex including the catalytic subunit of protein kinase A ($PKA_{cat}$), protein phosphatase 1 (PP1), *FKB-2*, and phosphodiesterase 4 (PDE-4) using mammalian anti-RyR, anti-PKA, anti-PP1, anti-calstabin, and anti-PDE-4 antibodies, respectively (*Figure 1E*). The previously published *C. elegans* anti-PDE-4 (*Charlie et al., 2006*) was used to detect PDE-4 on the channel. Our data show that *UNC-68* comprises a macromolecular complex, similar to that found in the mammalian muscle, that includes $PKA_{cat,}$ PP1, PDE-4, and FKB-2. *UNC-68* was depleted of FKB-2 in the FKB-2 (ok3007) null mutant (*Figure 1E and G*). In the FKB-2 null *C. elegans*, *UNC-68* and the rest of the macromolecular complex could not be immunoprecipitated using an anti-FKBP antibody (*Figure 1F and H*). Taken together these data indicate remarkable evolutionary conservation of the RyR macromolecular complex.

### Age-dependent biochemical and functional remodeling of UNC-68

RyR1 channels are oxidized, leaky, and $Ca^{2+}$ transients are reduced in aged mammalian skeletal muscle (*Andersson et al., 2011*). These changes occur by 2 years of age in mice (*Andersson et al., 2011*) and

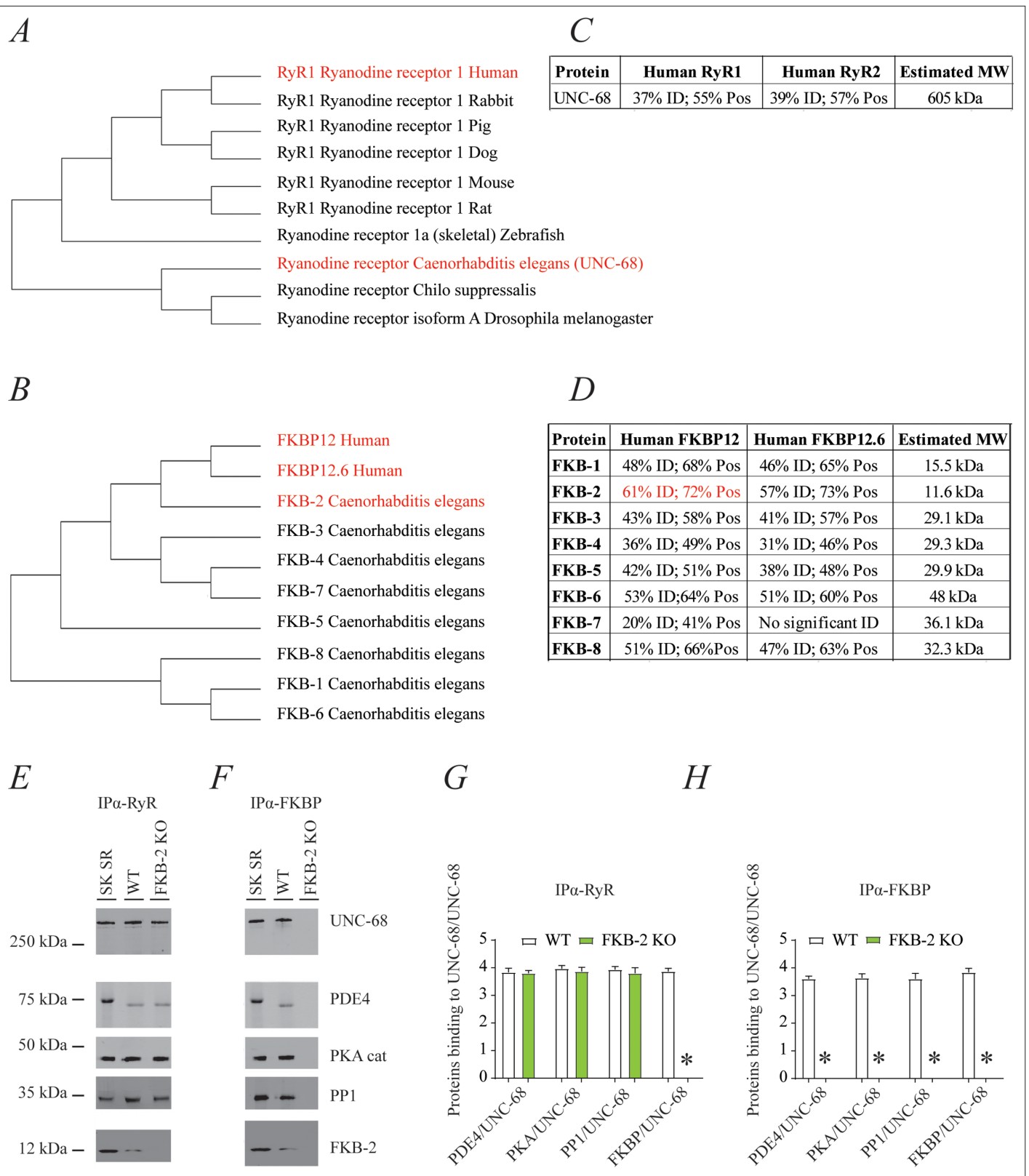

**Figure 1.** *UNC-68* comprises a macromolecular complex comparable to its mammalian homolog ryanodine receptor (RyR); RyR (**A**) and FKBP (**B**) evolution among species was inferred by the maximum likelihood method based on the JTT matrix-based model. (**C**) Homology comparison between *UNC-68* and the two human RyR isoforms (RyR1 and RyR2). (**D**) Homology comparison between the different FKB isoforms (1–8) and the human FKBP isoforms (FKBP12 and FKBP12.6). UNC-68 (**E**) and FKB-2 (**F**), respectively, were immunoprecipitated and immunoblotted using anti-RyR, anti-

*Figure 1 continued on next page*

*Figure 1 continued*

phosphodiesterase 4 (PDE4), anti-protein kinase A (catalytic subunit; PKA_{cat}), anti-protein phosphatase 1 (PP1), and anti-calstabin (FKBP) antibodies in murine skeletal sarcoplasmic reticulum preparations (SK SR), wild-type (WT) populations of *Caenorhabditis elegans*, and populations of *FKB-2 (ok3007)*. Images show representative immunoblots from triplicate experiments. (**G and H**) Quantification of bands intensity shown in E and F. Data are means ± SEM. One-way ANOVA shows * p<0.05 WT vs. FKB-2 KO. SK SR, sarcoplasmic reticulum fraction from mouse skeletal muscle. *Figure 1—source data 1*.

The online version of this article includes the following source data for figure 1:

**Source data 1.** Full incut gels of *Figure 1*.

by 80 years of age in humans. Similarly, *FKB-2* deficient worms exhibited an age-dependent decline in body wall muscle peak Ca$^{2+}$ transients starting at day 7 post-hatching (*Figure 2A–B*).

RyR1 oxidation has been linked to SR Ca$^{2+}$ leak and impaired muscle function during extreme exercise and in heart failure and muscular dystrophies (*Bellinger et al., 2008*; *Bellinger et al., 2009*; *Allen et al., 2008*). Furthermore, we have previously reported that oxidation of RyR1 and the subsequent intracellular Ca$^{2+}$ leak are underlying mechanisms of age-related loss of skeletal muscle specific force (force normalized to the cross-sectional area of muscle) (*Andersson et al., 2011*). WT *UNC-68* was oxidized (*Figure 2C–D*) and depleted of *FKB-2* (*Figure 2C–E*) and in an age-dependent manner. These changes mirror those occurring with extreme exercise in mice and humans (*Bellinger et al., 2008*) and in a murine model of Duchenne muscular dystrophy (*mdx* mice) characterized by impaired muscle function (*Bellinger et al., 2009*). Importantly, by 80 years of age, ~50% of humans develop severe muscle weakness that is a strong predictor of mortality due to falls, gait imbalance, and related factors (*Degens, 2007*). Similarly, *UNC-68* was significantly more oxidized (day 3–9) in *FKB-2 (ok3007)* worms compared to WT (*Figure 2C–D*).

To further demonstrate that *UNC-68* channels lacking *FKB-2* are inherently 'leaky', we used an assay that can monitor the rate of Ca$^{2+}$ released from the SR. Age synchronized worms' microsomes (day 5) were mixed with the Ca$^{2+}$ dye Fluo-4 and baseline fluorescence measurements were taken before adding 1 mM of ATP. By activating the sarco/endoplasmic calcium ATPase (SERCA) with ATP, cytosolic Ca$^{2+}$ is pumped into the microsomes, resulting in a decrease in Fluo-4 fluorescence. Once the fluorescence level plateaus, thapsigargin (SERCA antagonist) is added to block Ca$^{2+}$ reuptake into the SR. The rate at which the fluorescence increases directly correlates with the amount of Ca$^{2+}$ passively leaking into the cytoplasm: a higher increase of fluorescence compared to WT control indicates leaky *UNC-68* channels. Our data show that *UNC-68* from FKB-2 KO worms had a higher rate of SR Ca$^{2+}$ leak following thapsigargin administration compared to the WT channels (*Figure 2F*). This is corroborated by our previous findings, where disruption of RyR-calstabin binding increases the SR Ca$^{2+}$ leak in mammalian tissues (*Umanskaya et al., 2014*).

In mammals, calstabin regulation of RyR is tightly coupled to beta-adrenergic signaling (*Andersson et al., 2012*), and it is known that calstabin KO mice must undergo exercise stress before demonstrating a distinct muscle phenotype (*Bellinger et al., 2008*). Our method of inducing exercise stress in the worm was to place it in M9 buffer and observe it swimming, a well-described behavioral assay (*Lüersen et al., 2014*). By using an extended time trial of 2 hr, the worms fatigue and exhibit exercise-induced stress similar to that observed in mammals. Our data show a defect in *FKB-2 KO* swimming behavior over the course of its lifespan when compared to the WT. *FKB-2 KO* worms had decreased bending activity earlier in life, beginning at day 5, and an increased proportion of curling, a sign of fatigue (*Figure 2G–H*). Throughout midlife, the *FKB-2 KO* worms lag significantly behind their age-matched WT counterparts, suggestive of decreased muscle function. Furthermore, *FKB-2 KO* worms exhibit reduced lifespan compared to WT (*Figure 2I*).

## Pharmacologically mimicking aging phenotype affects Ca$^{2+}$ transient and impairs exercise capacity

*FKB-2* was competed off from the *UNC-68* macromolecular complex using rapamycin or FK506 (*Figure 3*). Both rapamycin and FK-506 bind to calstabin and compete it off from RyR channels, resulting in leaky channels and release of SR Ca$^{2+}$ in the resting state (*Kaftan et al., 1996*; *Tang et al., 2002*).

Age-synchronized young *C. elegans* (5 days) were treated with rapamycin or FK506. Ca$^{2+}$ transients were measured in partially immobilized transgenic nematodes expressing the genetically

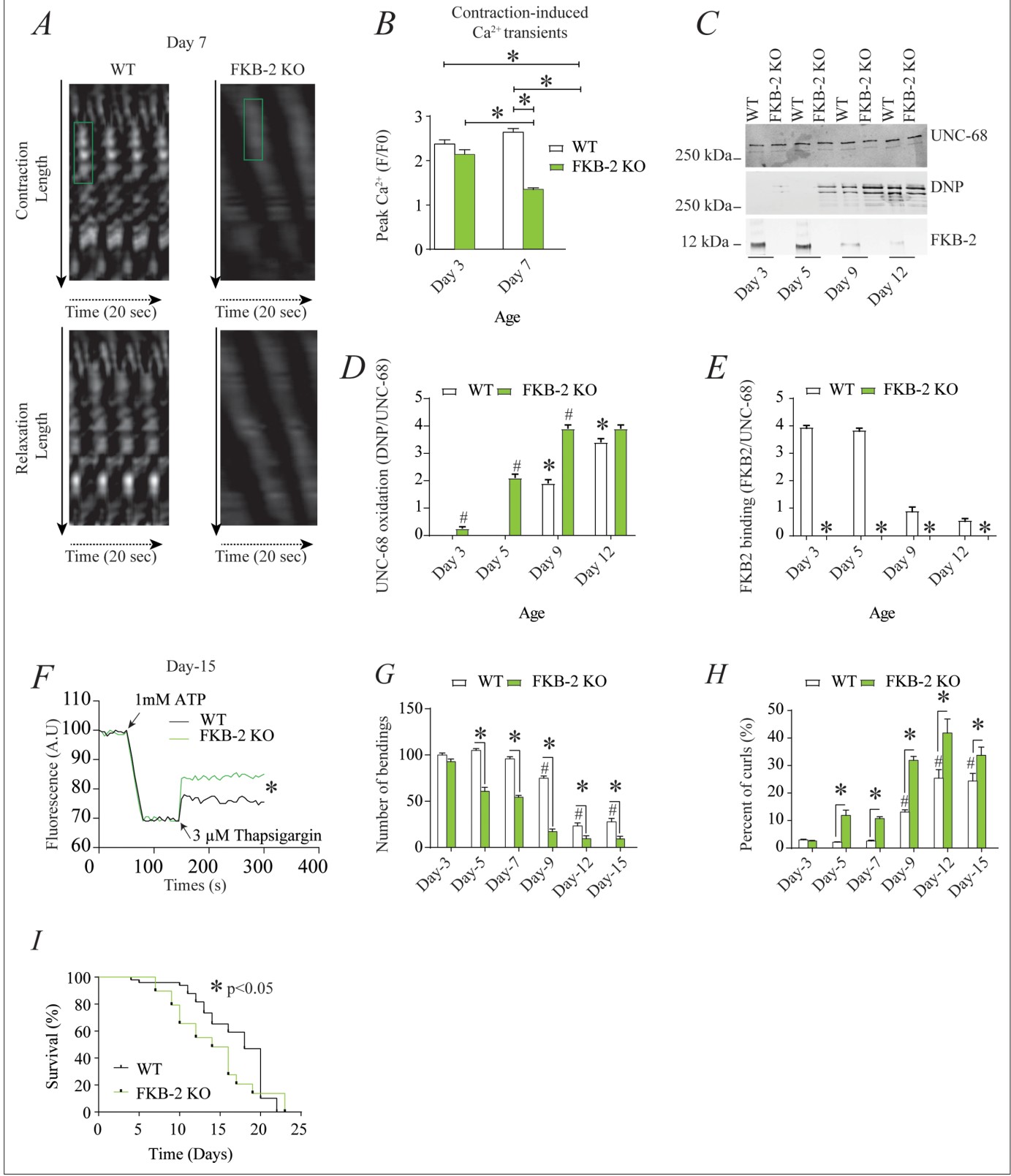

**Figure 2.** Remodeling of *UNC-68* and age-dependent reduction in intracellular calcium (Ca²⁺) transients is accelerated in FKB-2 *(ok3007)*
(**A**) Representative trace of Ca²⁺ transients from GCaMP2 wild type (WT) and FKB-2 KO (at day 7). Green box denotes peak fluorescence from worm's muscle during contraction. (**B**) Ca²⁺ transients in age-synchronized populations of WT and FKB-2 *(ok3007)* nematodes (at day 3 and 7); (**C**) *UNC-68* was immunoprecipitated from age-synchronized populations of mutant (FKB-2 KO) and WT nematodes (at day 3, 5, 9, and 12) and immunoblotted using

*Figure 2 continued on next page*

Figure 2 continued

anti-RyR, anti-calstabin, and dinitrophenyl (DNP; marker of oxidation) antibodies. (**D** and **E**) Quantification of the average band intensity from triplicate experiments: band intensity was defined as the ratio of each complex member's expression over its corresponding /*UNC-68's* expression. Data are means ± SEM. * p<0.05 WT vs. FKB-2 KO in panel D, # p<0.05 WT vs. FKB-2 KO in panel E, * p<0.05 WT at day 3 vs. WT at day 5 and day 9. (**F**) Ca$^{2+}$ leak assay performed with microsomes from WT and FKB-2 KO worms (day 5). Ca$^{2+}$ uptake into the microsomes was initiated by adding 1 mM of ATP. Then, 3 μM of thapsigargin was added to block the sarco/endoplasmic calcium ATPase activity. Increased fluorescence is proportional to the spontaneous Ca$^{2+}$ leakage throughout *UNC-68*. (**G**) Graph showing number of bends recorded for WT vs. FKB-2 KO worms at six distinct ages (day 3, 5, 7, 9, 12, and 15). (**H**) The number of curling events was calculated as a percentage of the overall motility (curls/bends). N = ~60 worms per group, except for day 15 (as fewer worms were alive at this timepoint). Day 15 = ~40 worms. (**I**) Percentage of survival of WT (average survival; 18 days) and FKB-2 KO worms (average survival; 14 days); Gehan-Breslow-Wilcoxon test for survival comparison was performed for statistical significance. Data are means ± SEM from triplicate experiments. One-way ANOVA shows * p<0.05 WT vs. FKB-2 KO, # p<0.05 WT at day 3 vs. WT at day 5, 7, 9, 12, and 15. *Figure 2—source data 1*.

The online version of this article includes the following source data for figure 2:

**Source data 1.** Full incut gels of *Figure 2*.

encoded Ca$^{2+}$ indicator, P*myo-3*::GCaMP2, in the body wall muscle cells (*Tallini et al., 2006*; *Liu et al., 2011*; *Figure 3A*). Pharmacologic depletion of *FKB-2* from *UNC-68* by rapamycin or FK506 treatment (15 min exposure to each drug) caused reduced body wall muscle Ca$^{2+}$ transients in WT *C. elegans* (*Figure 3B*). When FKB-2 was genetically depleted from the *UNC-68* complex, as in the *FKB-2* (*ok3007*) nematodes, treatment with rapamycin or FK506 had no effect on the Ca$^{2+}$ transients (*Figure 3B*).

Continuous Ca$^{2+}$ leak via *UNC-68* would be expected to result in depleted SR Ca$^{2+}$ stores; therefore, we utilized a common technique from the mammalian RyR literature to evaluate the SR Ca$^{2+}$ stores. In brief, an activating concentration of caffeine is used to fully open the RyR channel, leading to a rapid release of Ca$^{2+}$ from the SR into the cytoplasm. This increase can be approximated using a previously targeted, fluorescent Ca$^{2+}$ sensitive dye or indicator. Caffeine was applied to day 5 cut worms (*Figure 3C*), and the amount of fluorescence given off by GCaMP2 was measured. The GCaMP2-WT worms demonstrated a strong Ca$^{2+}$ transient within 10 s after caffeine administration, while GCaMP2-*FKB-2* KO worms failed to produce a response, suggesting that their SR Ca$^{2+}$ stores were too low to elicit one. Interestingly, GCaMP2-KO worms were observed as having very high background fluorescence, which may indicate an increase in cytosolic Ca$^{2+}$ from passive *UNC-68* leak.

Acute treatment with FK506 or rapamycin, for 15 min, each independently caused depletion of FKB-2 from the channel (*Figure 3D, E and F*) with no effect on the oxidation of UNC-68. Furthermore, longer treatment (2 and 4 hr) of WT worms with FK506 caused oxidation of *UNC-68*, demonstrating a relationship between depletion of *FKB-2* and oxidation of *UNC-68* (*Figure 3G,H,I*).

Indeed, rapamycin altered swimming behavior of WT but not *FKB-2* KO worms in a time-dependent manner (*Figure 3J*). Taken together with our Ca$^{2+}$ transient data, the observed muscle phenotype appears to be the result of *UNC-68* channel leak. These data suggest that rendering *UNC-68* channels leaky by removing *FKB-2* depletes SR Ca$^{2+}$, resulting in reduced Ca$^{2+}$ transients and weakened muscle contraction.

## Oxidation of UNC-68 causes reduced body wall muscle Ca$^{2+}$ transients

To investigate the individual effect of age-dependent *UNC-68* oxidation independent of the other confounding variables involved in aging (*Herndon et al., 2002*), we introduced a pharmacological intervention mimicking the aged state in young adult nematodes. Treating young adult nematodes (at 5 days of age) with the superoxide-generating agent paraquat (*Lee et al., 2003*) increased oxidation of *UNC-68* and depletion of *FKB-2* from the channel in a concentration-dependent manner (*Figure 4A, B and C*). Furthermore, contraction-associated Ca$^{2+}$ transients decreased with paraquat treatment in a concentration-dependent manner (*Figure 4D*). Indeed, treatment with antioxidant N-Acetyl-L-cysteine improved Ca$^{2+}$ transient in *FKB-2* KO worms (*Figure 4E*). These data indicate that both *UNC-68* oxidation and *FKB-2* depletion independently contribute to the observed aging body wall muscle deterioration.

To better clarify the role of oxidative stress in age-dependent *UNC-68* remodeling and Ca$^{2+}$ leak, we used two mutant mitochondrial electron transport chain (ETC) worms: the complex I mutant, CLK-1, and the complex II mutant, MEV-1. CLK-1 worms contain a Complex I-associated mutation such that they cannot synthesize their own ubiquinone (UQ), a redox active lipid that accepts and transfers

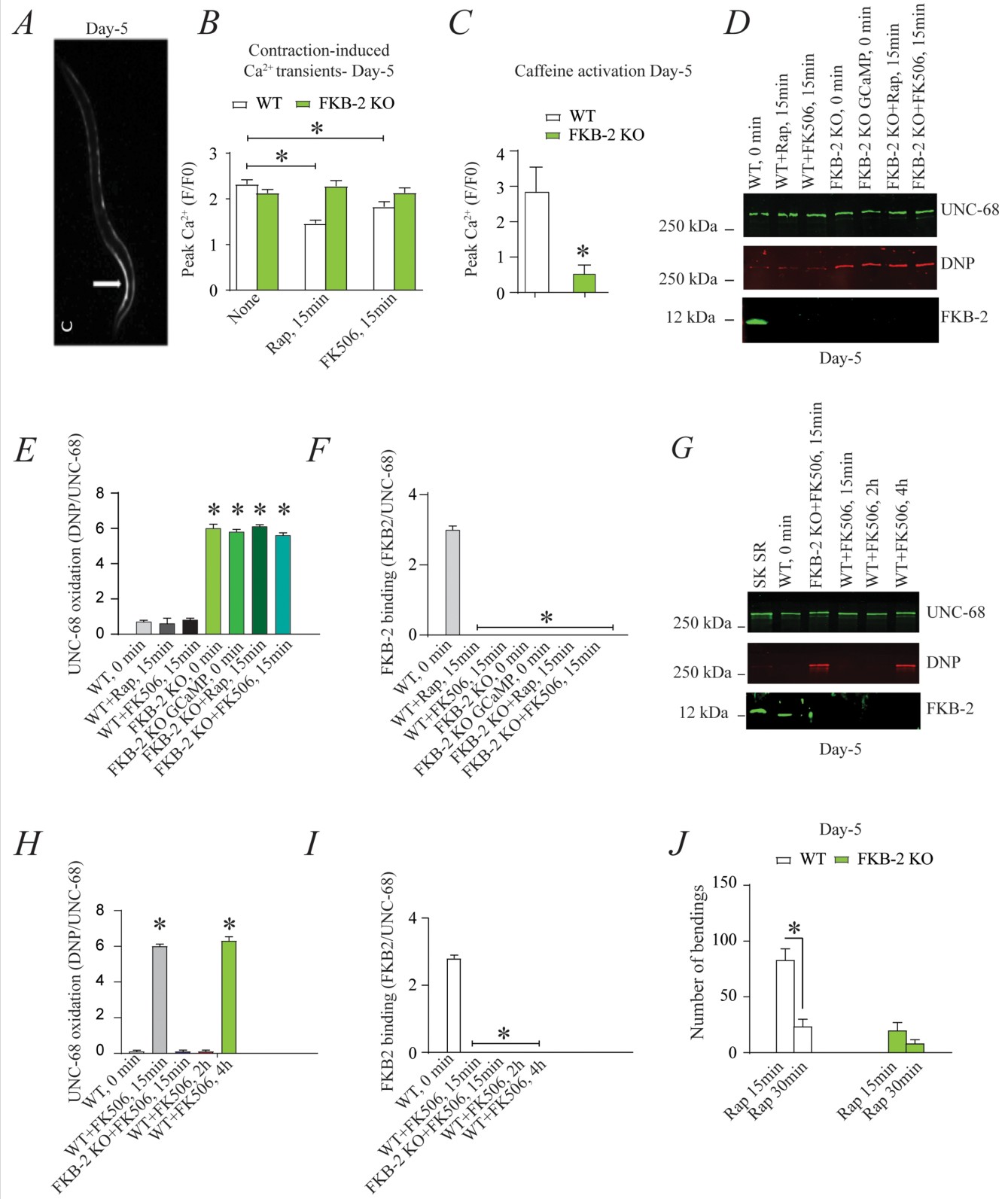

**Figure 3.** Depleting FKB-2 from *UNC-68* causes *UNC-68* oxidation (**A**) Representative image of caffeine activated calcium transient in GCaMP2 wild type (WT) at day 5; arrow denotes peak fluorescence in body wall muscle. (**B**) Intracellular calcium (Ca²⁺) transients in day 5 age-synchronized populations of WT and FKB-2 (*ok3007*) nematodes treated with 15 μM and 50 μM rapamycin and FK506, respectively (treatment was applied for 15 min). (**C**) Fluorescence intensity following caffeine activation in age-matched GCaMP2: WT vs. GCaMP2: FKB-2 KO worms at day 5. (**D**) *UNC-68* was

*Figure 3 continued on next page*

Figure 3 continued

immunoprecipitated and immunoblotted using anti-ryanodine receptor, anti-calstabin, and dinitrophenyl (DNP; marker of oxidation) antibodies in nematodes (at day 5) acutely treated with 15 μM and 50 μM rapamycin and FK506, respectively (treatment was applied for 15 min). (E–F) Quantification of the band intensity shown in (D): band intensity was defined as the ratio of either DNP (marker of *UNC-68* oxidation) or FKB-2 binding over its corresponding /*UNC-68's* expression. (G) *UNC-68* was immunoprecipitated after 0, 15 min, 2 hr, and 4 hr *FK506* exposure of the nematodes (at day 5). Representative immunoblots from triplicate experiments. (H–I) Quantification of the band intensity shown in (G): band intensity was defined as the ratio of either DNP (marker of *UNC-68* oxidation) or *FKB-2* binding over its corresponding /*UNC-68's* expression. (J) Graph showing number of bends recorded for WT vs. *FKB-2* KO worms (Day 5) treated for 20 and 30 min with 15 μM and 50 μM rapamycin and FK506, respectively. N ≥ 15 per group. Data are means ± SEM from triplicate experiments. One-way ANOVA shows * p<0.05 vs. WT for results shown in panel E, F, H, and I. Two-way ANOVA was used for results comparison in panel B, and t-test was used for results shown in C and J. SK SR; sarcoplasmic reticulum fraction from mouse skeletal muscle used as external control reference and was not quantified in the bar graphs. The time 0 min refers to untreated worms. *Figure 3—source data 1*.

The online version of this article includes the following source data for figure 3:

**Source data 1.** Full incut gels of *Figure 3*.

electrons from Complex I or II to Complex III in the ETC. The reduction in Complex I activity of CLK-1 is associated with long-lived worms (*Yang et al., 2011*; *Labuschagne et al., 2013*; *Kayser et al., 2004*). In contrast, MEV-1 worms contain a Complex II (succinate dehydrogenase) cytochrome B560 mutation (*Ishii et al., 1998*; *Senoo-Matsuda et al., 2001*; *Senoo-Matsuda et al., 2003*), preventing electron transfer from succinate to fumarate and causing mitochondrial ROS production, which is associated with decreased lifespan, averaging only 9 days (*Senoo-Matsuda et al., 2001*). Interestingly, we have seen increased *UNC-68* oxidation and *FKB-2* depletion in the short-lived mutant (MEV-1) compared to WT and long-lived mutant (CLK-1) worms (*Figure 4F, G and H*). Indeed, MEV-1 worms exhibited reduced exercise capacity compared to WT and CLK-1 worms (*Figure 4I–J*).

## UNC-68 Ca$^{2+}$ channel is a potential therapeutic target in aging

The small molecule Rycal S107 inhibits SR Ca$^{2+}$ leak by reducing the stress-induced depletion of calstabin from the RyR channel complex (*Bellinger et al., 2009*; *Lehnart et al., 2008*). Here, we show that treatment with S107 (10 μM) for 5 hr reassociated *FKB-2* with *UNC-68* without significant effect on the channel oxidation (*Figure 5A, B and C*). Furthermore, treatment with S107 improved peak Ca$^{2+}$ in an *FKB-2*-dependent manner, as demonstrated by the fact that treating the *FKB-2* KO worms did not change peak Ca$^{2+}$ (*Figure 5D–E*). Interestingly, S107 treatment reduced age-dependent impairment of exercise capacity in WT worms at day 15 (*Figure 5F*). Of note, S107 has no effect on the WT worms' lifespan (*Figure 5G*). Furthermore, the treatment of the short-lived worms, MEV1, with S107 restored the FKB-2 association with *UNC-68*, despite the persistence of the channel oxidation (*Figure 5H,I,J*).

## Discussion

Taken together, our data show that the *C. elegans* intracellular Ca$^{2+}$ release channel *UNC-68* comprises a macromolecular complex which is highly conserved throughout evolution from nematodes to humans. In nematodes, the *UNC-68* macromolecular complex is comprised of a similar array of regulatory subunits as the mammalian RyR1 channels: a phosphodiesterase PDE-4, a protein kinase PKA, a protein phosphatase PP1, and the immunophilin, *FKB-2*. Binding of *FKB-2* (the *C. elegans* homolog of the mammalian RyR stabilizing protein calstabin) to the *UNC-68* channel is required to prevent a pathological leak of intracellular Ca$^{2+}$, similar to the manner observed in mammalian muscle (*Andersson et al., 2011*). *C. elegans* exhibit reduced Ca$^{2+}$ transients, as well as oxidized *UNC-68 channels* and depleted *FKB-2* by ~2 weeks of age. Genetic *FKB-2* deficiency causes an accelerated aging phenotype; Ca$^{2+}$ transients are reduced in younger populations of *FKB-2 (ok3007)* nematodes and *UNC-68* is oxidized at an earlier time point in these mutants relative to WT. Treating aged WT nematodes with the RyR-stabilizing drug, S107, reassociates *FKB-2* with *UNC-68* and increases the Ca$^{2+}$ transients, indicating that *UNC-68* dysfunction is likely an underlying mechanism of age-dependent decrease in Ca$^{2+}$ transients in *C. elegans* body wall muscle. The mechanism causing age-dependent *UNC-68* dysfunction involves the loss of UNC-68/FKB-2 from the *UNC-68* channel complex due to oxidation of the channel. Of note, this may create a vicious cycle of intracellular Ca$^{2+}$ leak and oxidative overload in which leaky channels cause mitochondrial Ca$^{2+}$ accumulation and high levels of ROS production which

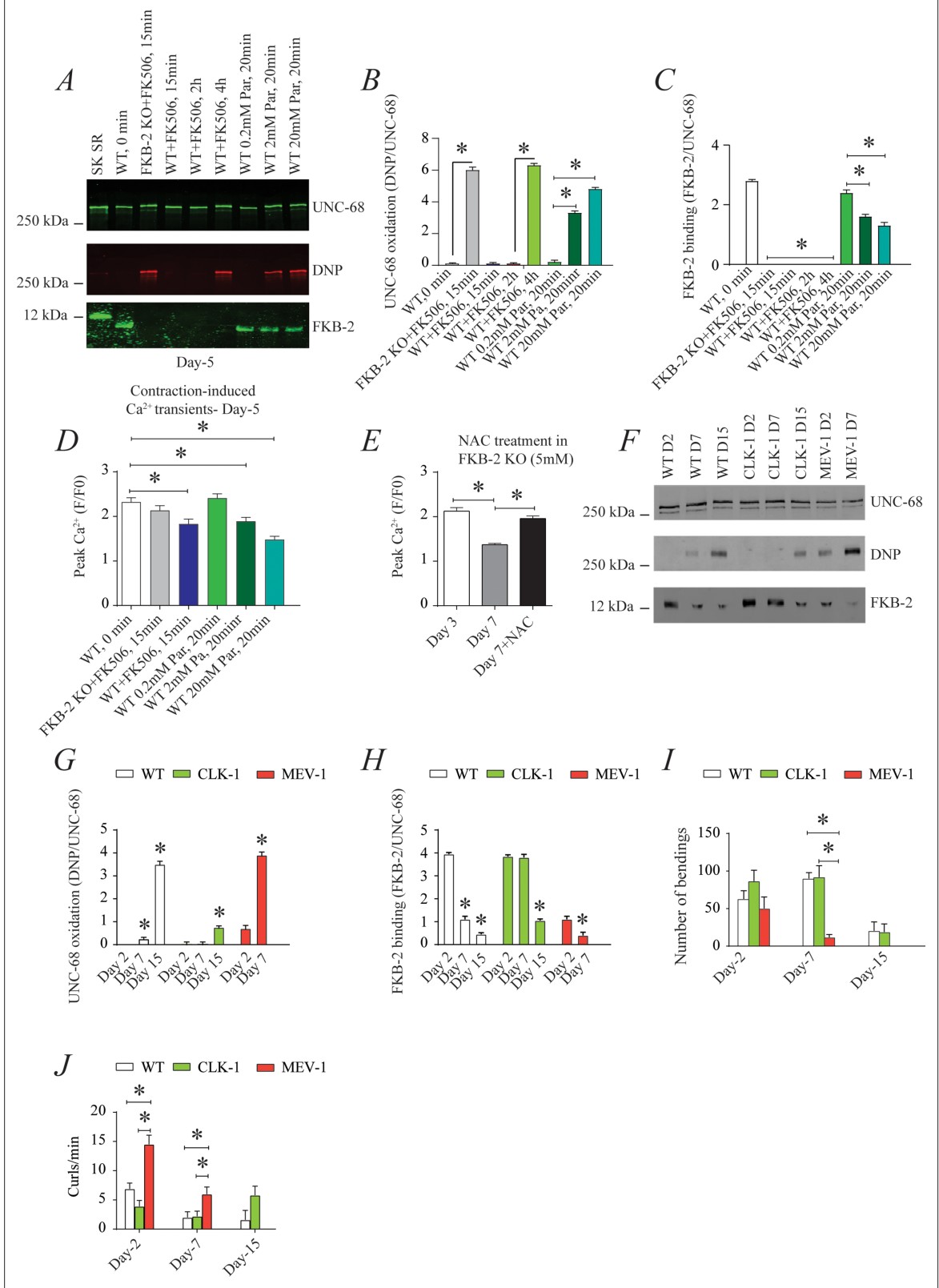

**Figure 4.** *UNC-68* oxidation causes defective intracellular calcium (Ca$^{2+}$) handling; (**A**) *UNC-68* was immunoprecipitated and immunoblotted using anti-ryanodine receptor (RyR), anti-calstabin, and dinitrophenyl (DNP; marker of oxidation) antibodies in nematodes acutely treated for 0, 15 min, 2 hr, or 4 hr with FK506 or paraquat (treatment was applied for 20 min) at increasing concentration (day 5). (**B–C**) Quantification of the band intensity shown in (**A**): band intensity was defined as the ratio of either DNP (marker of *UNC-68* oxidation) or FKB-2 binding over its corresponding /*UNC-68's*

*Figure 4 continued on next page*

*Figure 4 continued*

expression. (**D**) Contraction-associated Ca²⁺ transients measured in young age-synchronized WT nematodes treated for 15 with FK506 or for 20 min with increasing concentrations of paraquat (day 5). (**E**) Contraction-associated Ca²⁺ transients measured in FKB-2 KO nematodes treated with the antioxidant N-acetylcysteine (NAC) at 5 mM (day 7). (**F**) *UNC-68* was immunoprecipitated and immunoblotted using anti-RyR, anti-calstabin, and DNP (marker of oxidation) antibodies in WT, the long lived (CLK-1) and the short lived (MEV-1) nematodes at day 2, 7, and 15. (**G–H**) Quantification of the average band intensity from triplicate experiments: band intensity was defined as the ratio of each complex member's expression over its corresponding /*UNC-68's* expression. (**I**) Graph showing number of bends recorded for WT vs. CLK-1 and MEV-1 worms at three distinct ages (day 2, 7, and 15). (**J**) The number of curling events was calculated as a percentage of the overall motility (curls/bends). N ≥ 20 per group. Data are means ± SEM from triplicate experiments. One-way ANOVA shows * p<0.05. Two-way ANOVA was used in panel I and J. SK SR; sarcoplasmic reticulum fraction from mouse skeletal muscle used as external control reference and was not quantified in the bar graphs. The time 0 min refers to untreated worms. *Figure 4—source data 1*.

The online version of this article includes the following source data for figure 4:

**Source data 1.** Full incut gels of *Figure 4*.

further oxidize *UNC-68* and further exacerbate the Ca²⁺ leak over the course of the lifespan ( *Dridi et al., 2020a*) as has been demonstrated in mice (*Andersson et al., 2011*).

*C. elegans* exhibit an aging muscle phenotype similar to age-dependent loss of muscle function in humans (*Andersson et al., 2011*). This is characterized by impaired locomotion, reduction in muscle cell size associated with loss of cytoplasm and myofibrils, and progressive myofibril disorganization (*Herndon et al., 2002*). However, specific body wall muscle proteins involved in the *C. elegans* aging phenotype have not been determined. Here, we show that *UNC-68* is oxidized in aged nematodes and depleted of the channel-stabilizing protein, *FKB-2*. Our group has reported similar remodeling of RyR1 in skeletal muscle from aged mice (*Andersson et al., 2011*) and in murine models of muscular dystrophies (*Bellinger et al., 2009*), all of which exhibit intracellular Ca²⁺ leak and reduced muscle specific force production.

Though the oxidative stress theory of aging was first proposed in 1956 (*Hagen, 2003*; *Harman, 1956*), there is still substantial controversy surrounding the role of ROS in aging. For example, deletion or overexpression of the ROS detoxification enzyme superoxide dismutase has little effect on lifespan in *C. elegans* (*Gems and Doonan, 2009*; *Van Raamsdonk and Hekimi, 2012*). However, loss of *sesn-1*, the gene encoding sestrin, an evolutionarily conserved protein required for regenerating hyperoxidized forms of peroxiredoxins and for ROS clearance, causes reduced lifespan (*Yang et al., 2013*). Furthermore, ROS levels measured in vivo in *C. elegans* increase with age (*Back et al., 2012*). Other oxidative/antioxidative genes are involved in ROS production and may play a crucial role in the *UNC-68* oxidation (*Supplementary file 1*).

While the free radical theory of aging has taken a hit due to multiple observations that contradict the notion of a link between reduced oxidative load and longevity, the preponderance of data shows a correlation between oxidative damage and reduced lifespan (*Shields et al., 2021*). Moreover, there is no doubt that reduced muscle function is detrimental to survival (*Wilkinson et al., 2018*). The present study shows that a key effector of age-dependent oxidative overload, RyR1 channel leak and the resulting muscle dysfunction, occur approximately 2000 times faster in *C. elegans* compared to *Homo sapiens* and 50 times faster than in *Mus musculus*. Since the target system, RyR1/UNC68, is remarkably conserved and underlies dramatically similar physiological functions (namely SR Ca²⁺ release required for muscle contraction) the cause for the accelerated kinetics of aging must be determined elsewhere and in an unrelentingly constant manner as exemplified by the rigid control of species lifespan. There is however, only one known case of a significant prolongation of average lifespan in a species: *Homo sapiens*. Indeed, the average lifespan in the U.S. has doubled in the past century (*Schanzenbach et al., 2016*) largely due to improved sanitation and related public health measures that protect against communicable diseases, the present pandemic notwithstanding. This suggests that both environmental and intrinsic biological constraints can determine average lifespan. Since we are a species that can remodel our environment to a greater extent than others, we have been able to double our average lifespan by improving the environment, although now global warming threatens to reverse this achievement. The unanswered question remains what are the intrinsic biological constraints on a given species' longevity? Although, oxidative stress has been thought to be a major contributor to the skeletal muscle aging phenotype (*Andersson et al., 2011*), other biological factors, including changes to the nervous, hormonal, circulatory, and respiratory systems likely also play important roles.

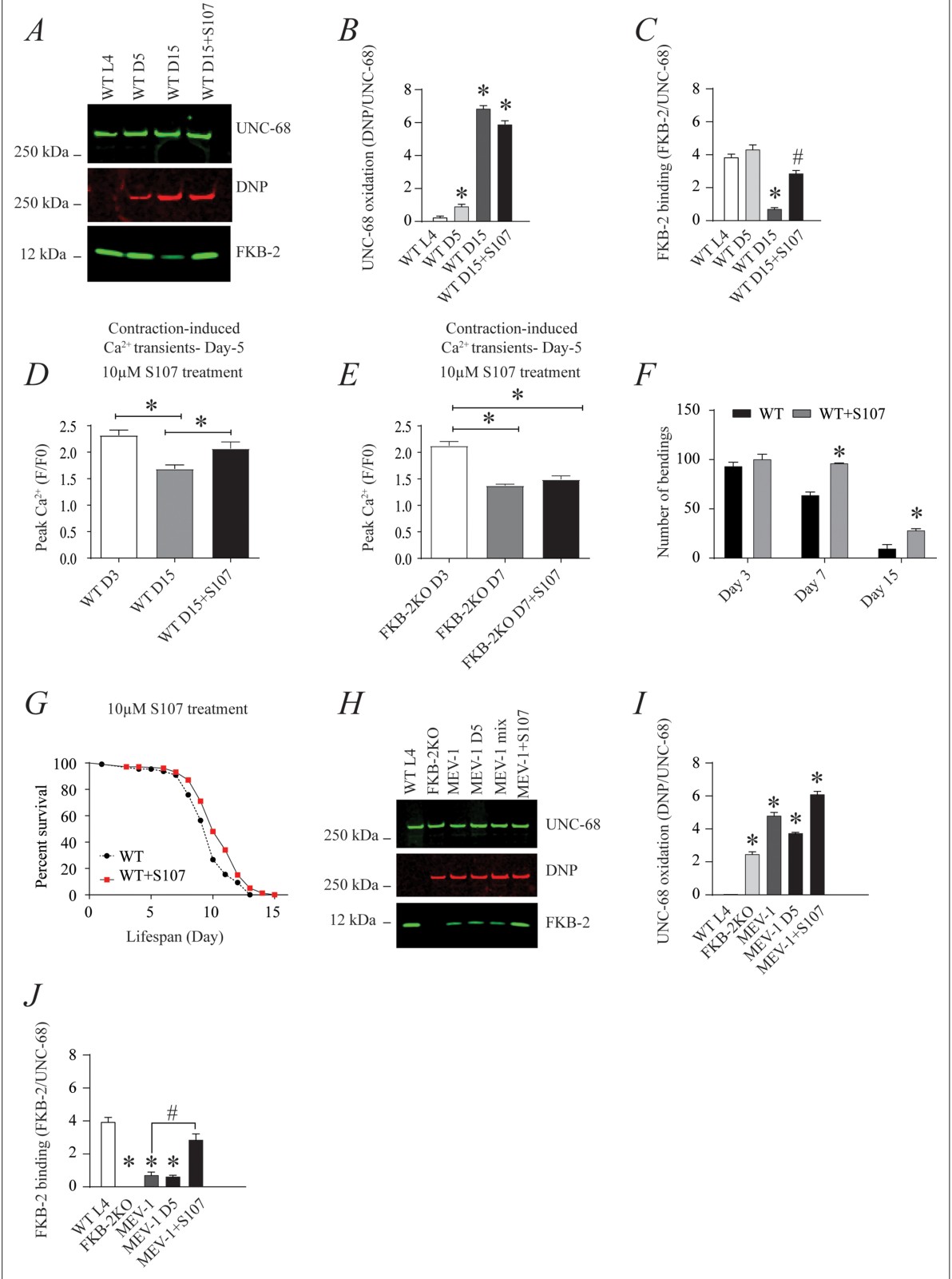

**Figure 5.** The ryanodine receptor (RyR)-stabilizing drug S107 increases body wall muscle calcium (Ca$^{2+}$) transients in aged *Caenorhabditis elegans*; (**A**) *UNC-68* was immunoprecipitated and immunoblotted with anti-RyR, anti-calstabin, and dinitrophenyl (DNP; marker of oxidation) in aged nematodes (Day L4, 5, and 15) with 10 μM of S107 (5 hr). (**B–C**) Quantification of the band intensity shown in (**A**): band intensity was defined as the ratio of either DNP (marker of *UNC-68* oxidation) or FKB-2 binding over its corresponding /*UNC-68* expression. Data are mean ± SEM. * p<0.05 vs. wild-typd L4

*Figure 5 continued on next page*

*Figure 5 continued*

(WTL4), # p<0.05 WT D15 vs. WT D15 + S107. (**D–E**) Contraction-associated $Ca^{2+}$ transients were measured in age-synchronized WT (day 3 and 15) (**D**) and (**E**) *FKB-2 KO worms* (day 3 and 7). Contraction-associated $Ca^{2+}$ transients in S107-treated worms were performed at day 15 for WT and day 7 for *FKB-2* worms. (**F**) Graph showing number of bends recorded for WT vs. WT treated with S107 worms at different ages (day 3, 7, and 15). (**G**) Percent of survival of WT vs. WT treated with S107 nematodes; Gehan-Breslow-Wilcoxon test for survival comparison was performed for statistical significance. (**H**) *UNC-68* was immunoprecipitated and immunoblotted with anti-RyR, anti-calstabin, and DNP (marker of oxidation) in short-lived nematodes (MEV-1) with S107 treatment (5 hr). (**I–J**) Quantification of the band intensity shown in (**H**): band intensity was defined as the ratio of either DNP (marker of *UNC-68* oxidation) or *FKB-2* binding over its corresponding /*UNC-68* expression. N ≥ 20 per group. Data are mean ± SEM from triplicate experiments. One-way ANOVA shows * p < 0.05 vs WT L4 unless otherwise indicated. In panel F, a t-test was used to compare WT and WT + S107 for each day. #p<0.05 MEV-1, vs. MEV-1 +S107 in panel J. *Figure 5—source data 1*.

The online version of this article includes the following source data for figure 5:

**Source data 1.** Full incut gels of *Figure 5*.

It would be interesting to know if the increased *UNC-68* oxidation-induced FKB-2 depletion and subsequent reduction in body wall muscle $Ca^{2+}$ transients are a result of globally increased ROS levels or increased ROS levels in *UNC-68*-surrounding microdomains. For example, we have previously shown that inducing RyR leak in enzymatically dissociated skeletal muscle cells causes increased mitochondrial membrane potential and mitochondrial ROS production (*Andersson et al., 2011*). Based on these data, we have proposed a model in which RyR1 leak (due to age-dependent oxidation of the channel and subsequent dissociation of calstabin) causes mitochondrial $Ca^{2+}$ overload, resulting in ROS production, thus leading to further oxidation of RyR1 and exacerbation of the SR $Ca^{2+}$ leak. This creates a vicious cycle between RyR1 and mitochondria that contributes to age-dependent loss of muscle function.

We also demonstrate that the putative null mutant, *FKB-2 (ok3007)*, prevents *FKB-2* from co-immunoprecipitating with *UNC-68*. The aging phenotype that we characterize in WT nematodes (biochemically modified *UNC-68* and reduced $Ca^{2+}$ transients) is accelerated in *FKB-2 (ok3007)*. There are eight *FKBs* that are homologous to mammalian calstabin in the *C. elegans* genome; *FKB-1* and *FKB-8* both have ~50% sequence identity to calstabin. Further studies could elucidate the possibility that in the absence of *FKB-2*, another *FKB* may stabilize *UNC-68*, in particular the aforementioned *FKB-8* (its gene is in close proximity to that of *FKB-2* on chromosome 2) and *FKB-1* (most similar to FKB-2 in terms of molecular weight). Such a mechanism could provide transitory compensation for the lack of FKB-2, in which other FKB isoform(s) bind to *UNC-68* with lower affinity. Because this binding is weak, and the channel is unstable, this compensation ends up failing at day 7 of age and the $Ca^{2+}$ leak is exacerbated. This hypothesis is partially supported by the unaltered $Ca^{2+}$ peaks in FKB2-KO worms at day 5 of age despite a complete depletion of FKB-2 binding protein. Such a compensatory mechanism was not observed with acute rapamycin and FK506 treatment potentially because, first, the $Ca^{2+}$ leak was acute and there was no time for a compensatory response, and, second, these drugs could act on all FKB isoforms.

Another key question is why *UNC-68* becomes oxidized within 2 weeks, whereas the same post-translational modification requires 2 years in mice and 80 years in humans (*Ljubuncic and Reznick, 2009*)? Given the high degree of conservation of RyR and other members of the complex (*Figure 1*), it is feasible that genetic screens in organisms such as *C. elegans* and *Drosophila* will yield additional crucial mediators that are common among species and explain disparities in age-dependent loss of muscle function such as genes-genes interactions, epigenetics or architecture, and gating of key proteins involved in aging such as RyR. Indeed, despite the conserved evolution of *UNC-68*, the channel contains higher numbers of methionine (3.5%) and serine (7.2%) (*Supplementary file 2*) compared to the human RyR1 (2.9 and 5.9%, respectively). Methionines are a primary target of oxidative stress that might cause defects in the channel gating and alter $Ca^{2+}$ release. Disparities in RyR1 serine residues among species, which are phosphorylated by protein kinases in response to stress, can cause conformational changes to the channel, exposing more residues to oxidation and could be a potential mechanism contributing to the accelerated *UNC-68* oxidation in *C. elegans*.

Regarding the conservation of EC coupling machinery, the *UNC-68* is localized to a specific portion of a vesicular network surrounding the myofilament lattice which suggests that the general architecture of the SR is conserved in metazoans. RyRs in vertebrate striated muscle cluster at internal couplings with Ttubules and, peripheral couplings adjacent to the surface membrane, visible as a 'feet'

in electron micrographs (*Maryon et al., 1998*). The 12–14 nm gap between the surface and SR vesicle membranes in *C. elegans* is identical to analogous gaps in vertebrate triad junctions suggesting that *UNC-68* bridges these gaps as seen in vertebrates. These similarities in muscle architecture further support our findings regarding the similar muscle aging phenotype between mammals and nematodes and the validity of *C. elegans* as a useful model to study age-dependent loss of muscle function.

Finally, *UNC-68* null mutants are defective in locomotion, but still propagate coordinated contraction waves by an unknown mechanism (*Maryon et al., 1996*). The only intracellular $Ca^{2+}$ release channels known in the SR vesicles, other than RyRs, are IP3Rs. The *C. elegans* genome contains a single IP3R gene, the *lfe-1* (or *itr-1*) (*Clandinin et al., 1998*). However, it has been reported that antibodies to *lfe-1* specifically stain the nerve ring but do not stain the myofilament lattice. Furthermore, *lfe-1* mutants exhibit normal motility suggesting the IP3Rs channels are not involved in the regulation of the body wall muscle contraction. Moreover, *UNC-68* has been reported to be expressed in neurons (*Sakube et al., 1997*) which may complicate the interpretation of its function in skeletal muscle. However, it seems that the neuronal expression is minor and does not modulate skeletal muscle function. Indeed, transformation of *UNC-68* null mutant animals with the WT *UNC-68* gene or the WT *UNC-68* coding sequence fused to the myo-3 promoter rescued motility defects and sensitivity to ryanodine-induced paralysis (*Maryon et al., 1998*). myo-3 is expressed in body wall muscles, as well as in enteric muscles (the enteric muscles do not affect motility). Furthermore, no staining of neurons has been observed with an anti-*UNC*-68 antibodies, which suggests that the major role of UNC-68 is supporting skeletal muscle contraction (*Maryon et al., 1998*).

Taken together, our data indicate that the *C. elegans* homolog of RyR, *UNC-68*, is comprised of a macromolecular complex and regulated by the immunophilin, *FKB-2*. We have identified age-dependent reduction in body wall muscle $Ca^{2+}$ transients in nematodes that is coupled to oxidation and remodeling of *UNC-68*. SR $Ca^{2+}$ stores are depleted in *FKB2*-KO worms, suggesting passive *UNC-68* leak. This observation is supported by the $Ca^{2+}$ leak assay results, which show that *FKB-2* regulation is critical in preventing *UNC-68* channels from aberrantly 'leaking' $Ca^{2+}$ into the cytoplasm. With reduced SR $Ca^{2+}$, *UNC-68* fails to release the burst of $Ca^{2+}$ required for normal E-C coupling, leading to impaired muscle function. Loss of muscle function is evident in the *FKB-2* KO worms during swimming trials, as middle-aged worms performed worse than their age-matched WT controls. Furthermore, our data strongly suggest a role for *FKB-2* and *UNC-68* in the age-dependent changes in $Ca^{2+}$ signaling, as treatment with the pharmacological RyR stabilizer S107 increases body wall muscle $Ca^{2+}$ transients. The advantage of targeting leaky RyR channels rather than using antioxidants would be the avoidance of the adverse effects of blocking beneficial oxidative signals.

# Materials and methods

## Key resources table

| Reagent type (species) or resource | Designation | Source or reference | Identifiers | Additional information |
|---|---|---|---|---|
| Strain, strain background (worms) | ok3007 | *Caenorhabditis* Genetics Center (University of Minnesota) | WormBase ID: WBVar00094093 | Genomic position: I: 2918075.12918967 |
| Strain, strain background (worms) | Pmyo-3:GCaMP2 worms | Kindly provided by Zhao-Wen Wang, University of Connecticut Health Center | | |
| Strain, strain background (worms) | mev-1 | *Caenorhabditis* Genetics Center (University of Minnesota) | WormBase ID: WBGene00003225 | Genomic position III: 10334277.10335168 |
| Strain, strain background (worms) | clk-1 | *Caenorhabditis* Genetics Center (University of Minnesota) | WormBase ID: WBGene00000536 | Genomic position III: 5277894.5279344 |
| Antibody | anti-RyR1 (Rabbit polyclonal) | Marks' lab, Columbia University, NY, USA | Cat. #: 5,029 Aa 1327–1339 | WB (1:1000), (10 µl) |
| Antibody | anti-PDE4 (Rabbit monoclonal) | Kindly provided by Kenneth Miller, Oklahoma Medical Research Foundation, Oklahoma City, Oklahoma | | WB (1:1000), (10 µl) |
| Antibody | anti-PP1 (Rabbit polyclonal) | Santa Cruz | Cat. #: sc6104 | WB (1:1000), (10 µl) |

*Continued on next page*

*Continued*

| Reagent type (species) or resource | Designation | Source or reference | Identifiers | Additional information |
|---|---|---|---|---|
| Antibody | anti-FKBP12 (Mouse monoclonal) | Santa Cruz | Cat. #: sc6104 | WB (1:2500), (10 µl) |
| Antibody | anti-FKBP12 (Rabbit polyclonal) | Abcam | Cat. #: ab2918 | WB (1:2000), (10 µl) |
| Commercial assay or kit | Oxyblot protein oxydation detection kit | Millipore | Cat. #: S7150 | WB (1:1000), (10 µl) |
| Chemical compound, drug | Rapamacin | Sigma Aldrich | Cat. #: 37,094 | |
| Chemical compound, drug | FK506 | Sigma Aldrich | Cat. #: Y0001926 | |
| Chemical compound, drug | Paraquat | Sigma Aldrich | Cat. #: 36,541 | |
| Chemical compound, drug | S107rycal drug | Marks' lab, Columbia University, NY, USA | | |
| Software, algorithm | GraphPad | GraphPad | V8.0 | |

## *C. elegans* strains and culture conditions

Worms were grown and maintained on standard nematode growth medium (NGM) plates on a layer of OP50 *Escherichia coli* at 20°C, as described (*Brenner, 1974*). N2 (Bristol) and *fkb-2 (ok3007)* were provided by the *Caenorhabditis* Genetics Center (University of Minnesota). *fkb-2 (ok3007)* was backcrossed six times. The transgenic strain expressing P*myo-3*: GCaMP2 was kindly provided by Zhao-Wen Wang, University of Connecticut Health Center (*Liu et al., 2011*). P*myo-3*: GCaMP2 was subsequently crossed into *fkb-2 (ok3007)* for measurement of contraction-associated Ca$^{2+}$ transients.

## Age synchronization

Adult worms at the egg-laying stage were treated with alkaline hypochlorite solution to obtain age-synchronized populations, and eggs were plated on NGM plates, as described (*Porta-de-la-Riva et al., 2012*). For experiments requiring aged worms, age-synchronized animals at the L4 stage were collected in M9 buffer and plated on NGM plates containing 5-fluoro-2'-deoxyuridine (FUDR, Sigma, 50 µM) to prevent egg-laying (*Mitchell et al., 1979*).

## Immunoprecipitation and immunoblotting

Nematodes were grown under standard conditions. For protein biochemistry experiments, a procedure to crack nematodes in a solubilizing and denaturing buffer was adapted (*Francis and Waterston, 1985*). Briefly, worms were washed and collected with M9 buffer, centrifuged for 2 min at 1000 rpm three times to wash. Worms were allowed to settle to the bottom of the collection tube by sitting on ice for ~5 min. Fluid was removed and the worm pellet was snap frozen in liquid nitrogen. Frozen pellets containing whole nematodes were rapidly thawed under warm running water. A volume of nematode solubilization buffer equal to the volume of the worm pellet was added (nematode solubilization buffer: 0.3% ethanolamine, 2 mM EDTA, 1 mM PMSF in DMSO, 5 mM DTT, 1× protease inhibitor), and tubes were microwaved (25 s for 100 µl pellet; time was increased for greater volumes). Lysates were then quickly drawn into a syringe through a 26-gauge needle and forced back through the needle into a new collection tube on ice. Samples were centrifuged at 1000 rpm for 2 min to remove insoluble material, and the supernatant was transferred to a new tube on ice. Lysates were snap frozen and stored in at –80°C.

A anti-mammalian RyR antibody (4 µg 5029 Ab [*Jayaraman et al., 1992*]) was used to immunoprecipitate *UNC-68* from 100 µg of nematode homogenate. Samples were incubated with the antibody in 0.5 ml of a modified RIPA buffer (50 mM Tris-HCl pH 7.4, 0.9% NaCl, 5.0 mM NaF, 1.0 mM Na3VO4, 1% Triton- X100, and protease inhibitors) for 1 hr at 4°C. The immune complexes were incubated with protein A Sepharose beads (Sigma, St. Louis, MS) at 4°C for 1 hr, after which time the beads were washed three times with buffer. Proteins were size-fractionated by SDS-PAGE (6% for *UNC-68*,

15% for FKB-2) and transferred onto nitrocellulose membranes for 1 hr at 200 mA (SemiDry transfer blot, Bio-Rad). After incubation with blocking solution (LICOR Biosciences, Lincoln NE) to prevent non-specific antibody binding, immunoblots were developed using antibodies against RyR (5029, 1:5000), PKAcat (Santa Cruz Biotechnology, sc-903, 1:1000), PDE4 (kindly provided to us by Kenneth Miller, Oklahoma Medical Research Foundation, Oklahoma City, Oklahoma), PP1 (sc6104, 1:1000), or an anti-calstabin antibody (Santa Cruz 1: 2500). To determine channel oxidation, the carbonyl groups on the protein side chains were derivatized to 2,4-dinitrophenylhydrazone (DNP-hydrazone) by reaction with 2,4-dinitrophenylhydrazine (DNPH) according to manufacturers (Millipore) instructions. The DNP signal on immunoprecipitated *UNC-68* was determined by immunoblotting with an anti-DNP antibody (Millipore, 1:1000). All immunoblots were developed and quantified using the Odyssey Infrared Imaging System (LICOR Biosystems, Lincoln, NE) and infrared-labeled secondary antibodies. In addition, immunoblotting and immunoprecipitation of the *UNC-68* macromolecular complex were conducted using another anti-calstabin antibody (1:2000, Abcam) and the same methods as described.

## Imaging contraction-associated body wall muscle Ca²⁺ transients

Spontaneous changes in body wall muscle $Ca^{2+}$ were measured in nematodes expressing GCaMP2 by fluorescence imaging using a Zeiss Axio Observer inverted microscope with an electron-multiplying CCD camera (Photometrics Evolve 512) and an LED light source (Colibri). Nematodes were partially immobilized by placing them individually into a 5–10 µl drop of M9 buffer, suspended between a glass slide and coverslip. 20-s videos of individual nematodes were recorded.

## Analyzing contraction-associated body wall muscle Ca²⁺ transients

Contraction-associated body wall muscle $Ca^{2+}$ transients were analyzed using an interactive data language-based image quantification software that was developed for this purpose in our laboratory. For each 20-s video, signals from the body wall muscles in nematodes expressing GCaMP2 fluorescence were analyzed using an edge-detection algorithm from each frame as 'line-scan' images, with the nematode perimeter on the y-axis and time (s) on the x-axis (*Xie et al., 2013*; *Yuan et al., 2014*). These images were then quantified based on the average of the peak $Ca^{2+}$ fluorescence signal on the worm muscle wall.

## Drug treatment

To pharmacologically deplete FKB-2 from *UNC-68*, nematodes were treated for 15 min with 15 µM rapamycin or imaging 50 µM FK506, respectively. To re-associate FKB-2 and *UNC-68*, aged nematodes were treated with 10 µM S107 for 3–5 hr. Oxidative stress was induced in the worms using 20 mM paraquat, a known generator of superoxide (*Wu et al., 2017*). Nematodes were grown in standard conditions, age-synchronized as described, washed and collected with M9 buffer, then centrifuged for 2 min at 1000 rpm three times. Worms were allowed to settle to the bottom of the collection tube by sitting on ice for ~5 min. Fluid was removed, the worm pellet was gently resuspended in M9 containing the appropriate drug concentration and gently rocked on a shaker at RT for the indicated time periods. Collection tubes were centrifuged for 2 min at 1000 rpm and M9 containing drug was removed and replaced with M9. Biochemistry or $Ca^{2+}$ measurements were then conducted as previously described (*Umanskaya et al., 2014*).

## Measuring SR Ca²⁺ stores using caffeine activation

Age-synchronized GCaMP2: WT and GCaMP2: FKB-2 KO were grown on NGM plates at 20°C they were separated from their progeny and left undisturbed until day 5. Individual worms were placed in a drop of M9 on a coverslip. The liquid was carefully wicked away using KIMTECH wipes until only a sliver of moisture surrounded the worm. The worm was quickly glued down to the coverslip using a tiny drop of DermaWorm applied to the head and tail of the worm before the worm desiccated. 80 µl of M9 buffer was added immediately afterward to polymerize the glue. Once the worm was secure, a clean lateral cut to the immediate tail region was made using a 20 G 1½ needle (adapted from Wang ZW et al., Neuron 2011[48]). An additional 170 µl of M9 buffer was applied for a total of 250 µl. The completed preparation was placed on the platform of a Zeiss confocal microscope; after 1 min at baseline, 25 mM of caffeine was added to an equal volume of M9 solution. The resulting body wall transients were recorded for 1 min.

## Calcium leak assay

Microsomes were prepared by centrifuging the *C. elegans* lysates (5 days synchronized populations) at 45,000× g for 30 min. Pellets were resuspended in lysis buffer containing 300 mM sucrose. Microsomes (5 µg/ml) were diluted into a 20 mM HEPES buffer (pH 7.2) containing 7 mM NaCl, 1.5 mM MgCl2, 120 mM K-gluconate, 5 mM K-phosphate, 8 mM K-phosphocreatine, 1 µM EGTA, and 2 µM CaCl2 mixed with 3 µM Fluo-4 and added to multiple wells of a 96-well plate. Calcium ($Ca^{2+}$) loading of the microsomes was initiated by adding 1 mM ATP. After $Ca^{2+}$ uptake and a new Fluo-4 signal baseline was observed, 3 µM Thapsigargin was added to inhibit the calcium uptake by the calcium pump (SERCA). The 'leak' of $Ca^{2+}$ out of the SR is measured by the increase in intensity of the Fluo-4 signal (measured in a Tecan infinite F500 fluorescence plate reader).

## Swimming behavior

Standard M9 buffer was mixed with 2% agar and poured into 96-well plates to create a planar surface for analyzing worm swimming behavior. Once the mixture had polymerized, approximately 180 µl of M9 was pipetted on top of the agar bed and age-synchronized worms from one of two groups (WT or FKB-2 KO) were placed individually into each well. To assess differences in exercise fatigue, worms were allowed to swim freely in M9 buffer for 2 hr; swimming bends and curls (*Lüersen et al., 2014*) were recorded by eye for 1 min. Representative videos were taken of each group, and investigators were blinded over the course of each experiment. All recordings were made in duplicate.

## Statistical analysis

All results are presented as mean ± SEM. Statistical analyses were performed using an unpaired two-tailed Student's t test (for two groups) and one-way ANOVA with Tukey-Kramer test (for three or more groups), unless otherwise indicated. For survival statistical comparison, we used Gehan-Breslow-Wilcoxon test. p-values <0.05 were considered significant. All statistical analyses were performed with Prism 8.0.

# Acknowledgements

This work was supported by grants from the NIH to ARM (T32HL120826, R01HL145473, R01DK118240, R01HL142903, R01HL061503, R01HL140934, R01AR070194).

# Additional information

### Competing interests

Andrew Marks: owns stock in ARMGO, Inc a company developing compounds targeting RyR and has patents on Rycals.US 2014/0378437, and US 7,718,644. The other authors declare that no competing interests exist.

### Funding

| Funder | Grant reference number | Author |
| --- | --- | --- |
| National Heart, Lung, and Blood Institute | R01HL145473 | Andrew Marks |
| National Institute of Diabetes and Digestive and Kidney Diseases | R01DK118240 | Andrew Marks |
| National Heart, Lung, and Blood Institute | R01HL142903 | Andrew Marks |
| National Heart, Lung, and Blood Institute | R01HL061503 | Andrew Marks |
| National Heart, Lung, and Blood Institute | R01HL140934 | Andrew Marks |

| Funder | Grant reference number | Author |
|---|---|---|
| National Institute of Arthritis and Musculoskeletal and Skin Diseases | R01AR070194 | Andrew Marks |
| National Heart, Lung, and Blood Institute | T32HL120826 | Andrew Marks |

The funders had no role in study design, data collection and interpretation, or the decision to submit the work for publication.

## Author contributions

Haikel Dridi, Conceptualization, Methodology, Writing – original draft, Writing – review and editing; Frances Forrester, Conceptualization, Data curation, Formal analysis, Methodology; Alisa Umanskaya, Formal analysis, Methodology; Wenjun Xie, Methodology; Steven Reiken, Conceptualization, Methodology, Project administration; Alain Lacampagne, Conceptualization, Data curation, Investigation, Methodology; Andrew Marks, Conceptualization, Data curation, Formal analysis, Funding acquisition, Investigation, Methodology, Project administration, Resources, Writing – original draft, Writing – review and editing

## Author ORCIDs

Haikel Dridi ⓘD http://orcid.org/0000-0001-9533-7367
Andrew Marks ⓘD http://orcid.org/0000-0002-8057-1502

## Decision letter and Author response

Decision letter https://doi.org/10.7554/eLife.75529.sa1
Author response https://doi.org/10.7554/eLife.75529.sa2

---

# Additional files

## Supplementary files

• Supplementary file 1. Oxidative regulators in *C. elegans* vs mice vs humans. Comparison of homologous oxidative and antioxidative genes between *C. elegans*, mouse and human. Criteria of comparison includes the function, the subcellular location, the enzymatic activity, mutation induces disrupted phenotype and percentage of homology.

• Supplementary file 2. Amino acid composition of UNC-68 and Human RyR1. Comparison of amino acid abundance in the *C. elegans* UNC-68 and the human RyR1 calcium channels. Number and percentage of Serines and methionines for each species are shown in red.

• Transparent reporting form

## Data availability

All data are described/available in the manuscript.

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
