## [Editor Report]

This manuscript will appeal to all with an interest in comparative physiology and the molecular biology of age-associated changes in muscle function. The authors draw parallels between aging skeletal muscle in humans and *C. elegans*, with evidence in support of age-dependent oxidation of the *C. elegans* ryanodine receptor ortholog, UNC-68, causing loss of the calstabin ortholog, FKB-2. This in turn results in calcium leak, reduced body wall calcium transients and muscle weakness, changes that are similar to those that occur in aging human skeletal muscle despite the dramatic differences in the lifespan of the two organisms.

---

## [Decision Letter]

**Decision letter after peer review:**

Thank you for submitting your article "Role of oxidation of excitation-contraction coupling machinery in age-dependent loss of muscle function in *C. elegans*" for consideration by *eLife*. Your article has been reviewed by 2 peer reviewers, one of whom is a member of our Board of Reviewing Editors, and the evaluation has been overseen by Richard Aldrich as the Senior Editor. The reviewers have opted to remain anonymous.

Essential revisions:

1. While researchers in the field may not be misled by the statement in the Introduction implying that muscle contraction is fully independent of extracellular ca^2+^, for the benefit of those who may not have a fully nuanced understanding of EC coupling in skeletal (or cardiac) muscle, the authors should modify the statement to include the concept of triggering-influxes of extracellular ca^2+^.

2. It is recommended that the authors sort out their thinking on the cause and effect relationships among UNC-68 oxidation, FKB-2 dissociation and UNC-68 leakiness, and present a cogent and compelling argument for them. It is not sufficient to present various lines of evidence showing that these things are interrelated and connected via a fuzzy mechanism to reduced ca^2+^ transients and an aging phenotype.

3. The authors should discuss differences in EC coupling in *C. elegans* relative to that of mammals and comment on the validity of *C. elegans* as a model for aging human muscle.

4. Please "dial back" your rhetoric on longevity.

5. Presentation is choppy. The presentation of all results shown in Figure 2 (Age-dependent…, FKB-2 deficiency… UNC-68 channels are leaky… and UNC-68 channel leak impairs…) should be merged into a single section, eliminating any redundancies in the process.

6. The strangely worded last sentence of the first paragraph of the "Pharmacologically mimicking aging phenotype…" section is internally illogical – the conclusion (second phrase) doesn't logically follow from the premise (first phrase). This muddles the message of this paragraph, which is presumably to convey the idea that it is the ability of rapamycin and FK506 to compete off calstabin (FKB-2 in this case), and not their well-known abilities to inhibit mTOR and calcineurin, respectively, that accounts for their observed effects on ca^2+^ transients, therefore justifying their use in mimicking the aging phenotype. The sentence should be revised to more accurately convey the authors' intended meaning.

7. Suggest organizing panels in Figure 3 to mirror organization in Figure 2 since they are similar data, just differing in "stimulus" (age in Figure 2 and pharmacology in Figure 3).

8. Need to clarify what time point the "WT+FK506" label corresponds to in all figure panels where it appears.

9. Nematode age should be indicated for all experiments/figure panels.

10. English quality could be improved (e.g., awkward phrasing, extraneously added or missing articles).

11. "This interaction [between RyR1 and CaV1.1] couples excitation of the sarcolemma to muscle contraction and eliminates dependence on extracellular ca^2+^." Direct coupling of RyR to sarcolemmal ca^2+^ channels via SPRY2 domains does not entirely eliminate the need for extracellular ca^2+^ for EC coupling; it merely reduces it to a very small influx necessary to trigger release from intracellular stores via a ca^2+^-induced ca^2+^-release mechanism.

12. Figure 2B:

– Is the day-15 bar for the FKB-2-KO missing? Or are there no ca^2+^ transients at this time point?

– Are these muscle contraction-induced transients or caffeine-induced transients?

13. FKB-2 deficiency accelerates age-dependent reduction in ca^2+^ transient that drives muscle contraction Not sure why this mini-section is here; recapitulates the first paragraph of the previous paragraph.

14. UNC-68 channels are leaky in the absence of FKB-2

"Once the fluorescence level plateaus…" Fluorescence level clearly has not plateaued in microsome samples from FKB-2-KO worms at the time thapsigargin is added. If the assay system is capable of truly reaching a plateau from which the resulting leak can be more accurately measured, experiments should be repeated; alternatively, a rationale should be provided for "jumping the gun" with thapsigargin addition.

15. UNC-68 channel leak impairs exercise capacity

"FKB-2 KO worms exhibit reduced life span compared to WT (Figure 2I)" – Maximum lifespan was not different; what are the average lifespans for WT and KO worms, and what statistical analysis supports the difference?

16. Pharmacologically mimicking aging phenotype affects ca^2+^ transient and impairs exercise capacity.

"…longer treatment of WT worms with FK506 caused oxidation of UNC-68, demonstrating a relationship between depletion of FKB-2 and oxidation of UNC-68" – This implies that depleting FKB-2 causes oxidation of UNC-68. This is causally backwards relative to the proposed model, where oxidation of UNC-68 (associated with aging) causes dissociation of FKB-2, leading to leaky channels and reduced muscle function and exercise capacity. Although in the Discussion, the authors do obliquely address the idea that dissociating FKB-2 could be expected to increase oxidation through increased UNC-68 leakiness, it comes too late in the presentation to do anything but muddle the message.

17. Figure 3F:

– The only way the labeling for this panel makes sense is if "WT+FK506" and "FKB-2 KO+FK506" labels are switched AND "WT+FK506" corresponds to a measurement at some earlier time point (0 or 1h? Not specified). Labeling in Figure 3D (and Figure 4C) is consistent with the idea that labels in 3F were switched, but the label "WT+FK506" is still ambiguous.

– Also, the "time course" is really a digital response that goes from ~0 at 2 h to max at 4 h. At minimum, data should be collected from a time point between 2 and 4 hours to demonstrate an actual time course.

18. Figure 4D:

– What is"WT+FK506" exactly? Is this the 4-h time point, as would make sense given that the comparison is with FKB-2 KO+FK506 (i.e., both conditions are associated with maximal UNC-68 oxidation), or is this a 0 h (or 1 h) time point, as suggested by the use of "WT+FK506" as the first term in the "WT+FK506", "WT+FK506, 2h" and "WT+FK506, 4h" series?

– Not clear why the WT+FK506 conditions would show a greater reduction in peak ca^2+^ than the FKB-2 KO condition (with or without FK506), since FKB-2 knockout presumably has a greater effect on UNC-68 subunit composition (i.e., FKB-2 completely gone) than competing off FKB-2 with FK506 (partially eliminated).

19. Figure 4A-D:

– Overall effects of FKB-2 KO/KD on ca^2+^ transients are not terribly impressive (Figure 4D) and not all that well correlated with UNC-68 oxidation (Figure 4C) or FKB-2 binding (Figure 4B). According to the authors' model, it is the loss of FKB-2 that ultimately is responsible for the leakiness of UNC-68 and associated phenotypes. Therefore, complete loss of FKB-2 binding should be more closely correlated with diminished ca^2+^ transients than UNC-68 oxidation (which presumably causes graded dissociation of FKB-2). By the same logic, FKB-2 KO should be more closely correlated with diminished ca^2+^ transients than WT+FK506 or paraquat (again, assuming that the paraquat-induced, UNC-68 oxidation-dependent dissociation of FKB-2 is also graded).

– The idea that the loss of FKB-2 is ultimately responsible for the leakiness of UNC-68 and associated phenotypes appeared to be the proposed model up until the presentation of data hinting that oxidation in and of itself was capable of producing this leakiness. In this context, only the paraquat FKB-2 binding/UNC-68 oxidation data were well correlated with peak ca^2+^ data.

20. Nematode age is a key parameter that is rarely specified in the text or figures/legends. Note that "young" is uninformative unless specifically defined.

21. "Another key question is why UNC-68 becomes oxidized within two weeks, whereas the same post-translational modification requires two years in mice and 80 years in humans (5)." This is not a key question. There are many, many contributors to the longevity of animals. It would be hard to find anyone who would argue that worms would live for years if only they had a greater antioxidant capacity.

22. The authors go to great lengths to connect UNC-68 oxidation with FKB-2 dissociation and link this relationship to UNC-68 leakiness and the aging phenotype. But it's never really clear whether it is the dissociation of FKB-2 or the oxidation of UNC-68 that underlies the leakiness. The authors clearly believe that FKB-2 dissociation is an important event in this sequence, otherwise, why use FKB-2-KO and WT+FK506 models throughout? But another formulation of their model (Discussion) is all oxidation all the time: "RyR1 leak (due to age-dependent oxidation of the channel) causes mitochondrial ca^2+^ overload, resulting in ROS production, thus leading to further oxidation of RyR1 and exacerbation of the SR ca^2+^ leak." Adding to this either/or quality is the apparent bidirectionality of the oxidation-FKB-2 dissociation relationship: not only does UNC-68 oxidation lead to FKB-2 dissociation, FKB-2 dissociation also leads to UNC-68 oxidation (presumably through feedback on mitochondrial ca^2+^ overload via increased UNC-68 leakiness). Lost in the presentation is how UNC-68 oxidation causes FKB-2 dissociation and whether (and how) UNC-68 oxidation causes increased UNC-68 leakiness in the absence of FKB-2 dissociation. All in all, the manuscript presents quite the Mobius strip with respect to cause-and-effect relationships.

---

## [Author Response]

Essential revisions:1. While researchers in the field may not be misled by the statement in the Introduction implying that muscle contraction is fully independent of extracellular ca^2+^, for the benefit of those who may not have a fully nuanced understanding of EC coupling in skeletal (or cardiac) muscle, the authors should modify the statement to include the concept of triggering-influxes of extracellular ca^2+^.

We thank the Reviewer for her/his comment. The three mammalian RyRs (RyR1, RyR2, RyR3) exhibit ca^2+^ induced ca^2+^ release (CICR), a process in which ca^2+^ itself activates the channel to release more ca^2+^. However, in physiological conditions extracellular ca^2+^ is not necessary for the activation of the mammalian skeletal muscle RyR1 to be activated. Indeed, ca^2+^ release from mammalian skeletal muscle, RyR1 isoform is essentially triggered by conformational changes to the voltage gated sensor in the DHPR channel, upon depolarization of the T-tubule (Rios and Pizarro 1991, Schneider 1994). This point of view has been supported by studies in which suppression of the extracellular ca^2+^ failed to blunt muscle contraction. Indeed, myofibers exhibit normal twitching even in a Ringer solution with 80 mM EGTA (Luttgau 1968). Ca^2+^ release and transients can be also recorded under voltage clamp at large positive potentials under which the slow inward ca^2+^ current becomes outwards (Brum 1987). Moreover, a series of mutagenesis studies have shown that the amino acid residues 489-503 in the DHPR-β1 subunit directly interact with RyR1 channels and mediate EC coupling (*Eltit et al., JBC, 2014)*. We believe that the Reviewer’s comment refers to the skeletal muscle EC coupling in amphibians (e.g., frog), which exhibit a minor dependence of extracellular ca^2+^ due to the expression of the β-RyR with similar gating properties to mammalian cardiac RyR2 (see Review Takashi 2011). We revised our introduction to clarify that we are describing the mechanism of EC coupling in mammals and in *C-elegans* (pages 2-3-4-5-6-7-8-10 and 12).

2. It is recommended that the authors sort out their thinking on the cause and effect relationships among UNC-68 oxidation, FKB-2 dissociation and UNC-68 leakiness, and present a cogent and compelling argument for them. It is not sufficient to present various lines of evidence showing that these things are interrelated and connected via a fuzzy mechanism to reduced ca^2+^ transients and an aging phenotype.

We thank the Reviewer for her/his comment which helped us to clarify our message. UNC-68 oxidation causes depletion of FKB-2 from the channel, which results in ca^2+^ leak. This ca^2+^ leak leads to excessive ca^2+^ accumulation in the mitochondria, generates ROS and further oxidizes UNC-68. In disease conditions (e.g. aging), this mechanism results in a vicious cycle in which oxidation promotes FKB-2 depletion which subsequently promotes further oxidation of UNC-68 or in mammalian muscle, RyR1 (Ref Andersson et al., Mol Cell). We revised our manuscript to better clarify this mechanism (Last paragraph, page 10).

3. The authors should discuss differences in EC coupling in *C. elegans* relative to that of mammals and comment on the validity of *C. elegans* as a model for aging human muscle.

We thank the Reviewer for his comment which helped improve the manuscript. We have discussed the differences in EC coupling in *C. elegans* vs. mammals and commented on the validity of *C. elegans* as a model for aging human muscle (see revised discussion-page 13).

4. Please "dial back" your rhetoric on longevity.

We have largely replaced the word “lifespan” by “age-dependent loss of muscle function” throughout the revised manuscript.

5. Presentation is choppy. The presentation of all results shown in Figure 2 (Age-dependent…, FKB-2 deficiency… UNC-68 channels are leaky… and UNC-68 channel leak impairs…) should be merged into a single section, eliminating any redundancies in the process.

We thank the Reviewer for her/his comment which helped achieve a better organization of the manuscript. In the revised version, we merged the presentation of the results shown in Figure 2 into 1 section and eliminated the redundancies (pages 6 and 7).

6. The strangely worded last sentence of the first paragraph of the "Pharmacologically mimicking aging phenotype…" section is internally illogical – the conclusion (second phrase) doesn't logically follow from the premise (first phrase). This muddles the message of this paragraph, which is presumably to convey the idea that it is the ability of rapamycin and FK506 to compete off calstabin (FKB-2 in this case), and not their well-known abilities to inhibit mTOR and calcineurin, respectively, that accounts for their observed effects on ca^2+^ transients, therefore justifying their use in mimicking the aging phenotype. The sentence should be revised to more accurately convey the authors' intended meaning.

We thank the Reviewer for her/his comment which helped clarifying our message on the use of FK506 and rapamycin to compete off FKB-2. We revised these statement in the new version of the manuscript and added references supporting the significance of these experiments (Last paragraph, page 7).

7. Suggest organizing panels in Figure 3 to mirror organization in Figure 2 since they are similar data, just differing in "stimulus" (age in Figure 2 and pharmacology in Figure 3).

In the revised manuscript, the panels organization in Figure 3 mirrors the panel organization in Figure 2.

8. Need to clarify what time point the "WT+FK506" label corresponds to in all figure panels where it appears.

We apologize for not adding this important information. “WT+FK506” label correspond to a short treatment (15min) of WT worms with FK506 drug. We have revised all our figures and figure legends to show the time point for each experiment.

9. Nematode age should be indicated for all experiments/figure panels.

Again, we thank the Reviewer for her/his comment we apologize for not adding this important information in some experiments. We added the nematode age for all the experiments described in the manuscript.

10. English quality could be improved (e.g., awkward phrasing, extraneously added or missing articles).

We revised our manuscript to improve the English quality and remove potentially confusing words.

11. "This interaction [between RyR1 and CaV1.1] couples excitation of the sarcolemma to muscle contraction and eliminates dependence on extracellular ca^2+^." Direct coupling of RyR to sarcolemmal ca^2+^ channels via SPRY2 domains does not entirely eliminate the need for extracellular ca^2+^ for EC coupling; it merely reduces it to a very small influx necessary to trigger release from intracellular stores via a ca^2+^-induced ca^2+^-release mechanism.

We thank the Reviewer for her/his remark. We agree that the physical interaction between RyR1 and Cav_1.1_ does not eliminate the need for extracellular ca^2+^ or the activation of RyR1 by ca^2+^ but it renders it unnecessary to achieve a normal contraction. Please refer to the answer to question 1 for details. In the revised manuscript, we replace “eliminates dependence on extracellular ca^2+^” by “reduces dependence on extracellular ca^2+^” (Page 4)

12. Figure 2B:– Is the day-15 bar for the FKB-2-KO missing? Or are there no ca^2+^ transients at this time point?– Are these muscle contraction-induced transients or caffeine-induced transients?

We thank the Reviewer for her/his comment, and we apologize for not being clear enough in the description of Figure 2B. Indeed, we did not measure ca^2+^ in FKB-2-KO worms at day 15 as the mutant already exhibited a significant reduction in ca^2+^ peak at day 7. The ca^2+^ transient at day 15 was only measured in WT since their ca^2+^ peak was not affected at day 7. To be consistent, we removed the WT Day 15 ca^2+^ transient. We also indicated on the top of panel 2B whether these measurements are contraction-induced ca^2+^ transients or caffeine-induced ca^2+^ transients.

13. FKB-2 deficiency accelerates age-dependent reduction in ca^2+^ transient that drives muscle contraction Not sure why this mini-section is here; recapitulates the first paragraph of the previous paragraph.

We have removed this redundancy in the revised version of the manuscript.

14. UNC-68 channels are leaky in the absence of FKB-2"Once the fluorescence level plateaus…" Fluorescence level clearly has not plateaued in microsome samples from FKB-2-KO worms at the time thapsigargin is added. If the assay system is capable of truly reaching a plateau from which the resulting leak can be more accurately measured, experiments should be repeated; alternatively, a rationale should be provided for "jumping the gun" with thapsigargin addition.

We have repeated this leak assay experiment and adjusted the conditions, mainly by using a higher amount of microsomes, which allowed us to reach a fluorescence plateau after adding thapsigargin. The new results are displayed in the new Figure 2F.

15. UNC-68 channel leak impairs exercise capacity"FKB-2 KO worms exhibit reduced life span compared to WT (Figure 2I)" – Maximum lifespan was not different; what are the average lifespans for WT and KO worms, and what statistical analysis supports the difference?

The median WT survival was 18 days versus 14 days for FKB-2-KO worms. We performed Gehan-Breslow-Wilcoxon test for survival comparison which gives more weight to death at early time points. Since we do not have censored worms at any time point, we chose this test over the log-rank test to take in consideration the early death of worms. We added these details to the figure 2 legend.

16. Pharmacologically mimicking aging phenotype affects ca^2+^ transient and impairs exercise capacity."…longer treatment of WT worms with FK506 caused oxidation of UNC-68, demonstrating a relationship between depletion of FKB-2 and oxidation of UNC-68" – This implies that depleting FKB-2 causes oxidation of UNC-68. This is causally backwards relative to the proposed model, where oxidation of UNC-68 (associated with aging) causes dissociation of FKB-2, leading to leaky channels and reduced muscle function and exercise capacity. Although in the Discussion, the authors do obliquely address the idea that dissociating FKB-2 could be expected to increase oxidation through increased UNC-68 leakiness, it comes too late in the presentation to do anything but muddle the message.

We addressed the concept of “oxidation-ca^2+^ leak” vicious cycle earlier in the discussion (Page 10).

17. Figure 3F:– The only way the labeling for this panel makes sense is if "WT+FK506" and "FKB-2 KO+FK506" labels are switched AND "WT+FK506" corresponds to a measurement at some earlier time point (0 or 1h? Not specified). Labeling in Figure 3D (and Figure 4C) is consistent with the idea that labels in 3F were switched, but the label "WT+FK506" is still ambiguous.

We thank the Reviewer for her/his comment, and we apologize for the lack of clarity. Indeed, WT+FK506 are WT worms treated with FK506 for 15 min. The worms exhibit complete depletion of FKB-2 but no oxidation. The oxidation only appears when the WT worms are treated with FK506 for 4 hours. The FKB-2 KO worms have basal oxidation independent of FK506 treatment (no difference between FKB-2-KO treated and untreated), (see revised Figure 3, Panel D and G). The basal oxidation in the FKB-2 KO worms could be due to constitutive ca^2+^ leak which mitochondrial ca^2+^ overload and oxidative stress in this strain.

The first line of the gel is not quantified (no bar graphs are shown), it represents an immunoprecipitated RyR1 from mouse skeletal muscle and serves as a positive control for the antibody. The quantification bars reflect only the experiments performed with worms. We revised the figure 3, and 4 legends to better clarify these details.

– Also, the "time course" is really a digital response that goes from ~0 at 2 h to max at 4 h. At minimum, data should be collected from a time point between 2 and 4 hours to demonstrate an actual time course.

We thank the Reviewer for her/his comment, and we apologize for the lack of clarity regarding the time course experiments. Indeed, our time course includes 0, 15 min, 2 h and 4 h. We have added these missing details to the figure and the legend.

18. Figure 4D:– What is"WT+FK506" exactly? Is this the 4-h time point, as would make sense given that the comparison is with FKB-2 KO+FK506 (i.e., both conditions are associated with maximal UNC-68 oxidation), or is this a 0 h (or 1 h) time point, as suggested by the use of "WT+FK506" as the first term in the "WT+FK506", "WT+FK506, 2h" and "WT+FK506, 4h" series?

“WT+FK506” is WT worms treated with FK506 for 15 min. This group exhibits complete depletion of FKB-2 but no oxidation. Of note, the first line of the gel is not quantified (no bar graphs are shown), it represents an immunoprecipitated RyR1 from mouse skeletal muscle and serves as a positive control for the antibody. This detail has been added to the figure 4 legend.

– Not clear why the WT+FK506 conditions would show a greater reduction in peak ca^2+^ than the FKB-2 KO condition (with or without FK506), since FKB-2 knockout presumably has a greater effect on UNC-68 subunit composition (i.e., FKB-2 completely gone) than competing off FKB-2 with FK506 (partially eliminated).

We thank the Reviewer for her/his very interesting comment. We agree with the Reviewer that the FKB-2-KO worms were expected to have a greater reduction in peak ca^2+^ compared to WT+FK506 at any investigated age. However, this was not the case at day 5. One possibility is that the FKB2-KO worms develop a compensatory mechanism over time due to the chronic ca^2+^ leak to protect the nematode from severe pathogenic ca^2+^ overload at an early age. This mechanism ends up failing over time and the ca^2+^ leak is exacerbated around day 7 of age. However, the adaptation to chronic ca^2+^ leak is an important question that needs further investigation in future and is beyond the scope in the current study. Please see discussion, first paragraph of page 13 for proposed potential mechanisms.

19. Figure 4A-D:– Overall effects of FKB-2 KO/KD on ca^2+^ transients are not terribly impressive (Figure 4D) and not all that well correlated with UNC-68 oxidation (Figure 4C) or FKB-2 binding (Figure 4B). According to the authors' model, it is the loss of FKB-2 that ultimately is responsible for the leakiness of UNC-68 and associated phenotypes. Therefore, complete loss of FKB-2 binding should be more closely correlated with diminished ca^2+^ transients than UNC-68 oxidation (which presumably causes graded dissociation of FKB-2). By the same logic, FKB-2 KO should be more closely correlated with diminished ca^2+^ transients than WT+FK506 or paraquat (again, assuming that the paraquat-induced, UNC-68 oxidation-dependent dissociation of FKB-2 is also graded).

We thank the Reviewer for her/his comment which help clarifying the results displayed in Figure 4A-D. These experiments were done at 5 days of age. Despite a significant UNC-68 oxidation and absence of FKB-2, the FKB-2-KO worms do not exhibit significant reduction in ca^2+^ transients until day 7. Again, as stated in response to question 18, this could be due to a compensatory mechanism by which these worms overcome the chronic ca^2+^ leak early in life but ends up failing at 7 days. In contrast, the WT worms do not develop a compensatory mechanism upon acute oxidation (paraquat treatment) or FKB-2 depletion (FK506 treatment) and therefore rapidly exhibit a leaky ca^2+^ phenotype (reduced ca^2+^ transient). We discussed these details and added the age of worms for each experiment in the revised version of the manuscript as requested by the reviewer (pages 13).

– The idea that the loss of FKB-2 is ultimately responsible for the leakiness of UNC-68 and associated phenotypes appeared to be the proposed model up until the presentation of data hinting that oxidation in and of itself was capable of producing this leakiness. In this context, only the paraquat FKB-2 binding/UNC-68 oxidation data were well correlated with peak ca^2+^ data.

The FKB-2 KO worms do not exhibit a reduction in the ca^2+^ transient at 5 days of age. See answer to question 18 and 19. In WT worms, paraquat treatment causes acute UNC-68 oxidation and subsequently depletes FKB-2 of channel resulting in an acute ca^2+^. Under these conditions, the WT worms have no time to adapt their ca^2+^ machinery and as such exhibit a more severe ca^2+^ leak phenotype at this age (day 5). Again, the pharmacological depletion of FKB-2 is acute, and its consequences are rapidly exacerbated. We discussed these aspects in the revised manuscript (page 13).

20. Nematode age is a key parameter that is rarely specified in the text or figures/legends. Note that "young" is uninformative unless specifically defined.

We added the specific age for each experiment in the revised text and in the figures/legends.

21. "Another key question is why UNC-68 becomes oxidized within two weeks, whereas the same post-translational modification requires two years in mice and 80 years in humans (5)." This is not a key question. There are many, many contributors to the longevity of animals. It would be hard to find anyone who would argue that worms would live for years if only they had a greater antioxidant capacity.

We think this is an important question because the enzymes involved in UNC-68 oxidation are very similar to those causing age-dependent oxidation of RyR1 in mammals which takes orders of magnitude longer. So, if one could answer this question, it would shed some light on the unanswered question of what determines species-specific differences in lifespan. The point is not that the oxidation is the most important issue, but rather the etiology of the tremendous difference in kinetics of the age-dependent signals between species. Despite all of the research on aging this question has yet to be answered.

22. The authors go to great lengths to connect UNC-68 oxidation with FKB-2 dissociation and link this relationship to UNC-68 leakiness and the aging phenotype. But it's never really clear whether it is the dissociation of FKB-2 or the oxidation of UNC-68 that underlies the leakiness. The authors clearly believe that FKB-2 dissociation is an important event in this sequence, otherwise, why use FKB-2-KO and WT+FK506 models throughout? But another formulation of their model (Discussion) is all oxidation all the time: "RyR1 leak (due to age-dependent oxidation of the channel) causes mitochondrial ca^2+^ overload, resulting in ROS production, thus leading to further oxidation of RyR1 and exacerbation of the SR ca^2+^ leak." Adding to this either/or quality is the apparent bidirectionality of the oxidation-FKB-2 dissociation relationship: not only does UNC-68 oxidation lead to FKB-2 dissociation, FKB-2 dissociation also leads to UNC-68 oxidation (presumably through feedback on mitochondrial ca^2+^ overload via increased UNC-68 leakiness). Lost in the presentation is how UNC-68 oxidation causes FKB-2 dissociation and whether (and how) UNC-68 oxidation causes increased UNC-68 leakiness in the absence of FKB-2 dissociation. All in all, the manuscript presents quite the Mobius strip with respect to cause-and-effect relationships.

Oxidation of UNC-68 causes a loss of FKB-2 from the channel. The precise mechanism by which this occurs is unknown and the subject of ongoing structural studies in our laboratory. The result is leaky UNC-68 channels. The same leak can be observed by removing FKB-2 without oxidation (e.g with FK506 or in the FKB-2 KO worms). Conversely, preventing FKB-2 removal with a Rycal drug (S107) can prevent leak even when the channels are oxidized. Thus, we conclude that it is the loss of FKB-2 from the channel that destabilizes the closed state and causes leak, not the oxidation by itself. The oxidation becomes important in the context of lifespan because it is the age-dependent increase in oxidative load in the worms that ultimately leads to UNC-68 oxidation in vivo.